# Spatiotemporal denudation rates of the Swabian Alb escarpment (Southwest Germany) dominated by anthropogenic impact, lithology, and base-level lowering

Mirjam Schaller[1,2], Daniel Peifer[1], Alexander B. Neely[1], Thomas Bernard[1], Christoph Glotzbach[1], Alexander R. Beer[1], Todd A. Ehlers[2]

[1]Department of Geosciences, University of Tuebingen, 72076 Tuebingen, Germany
[2]School of Geographical and Earth Sciences, University of Glasgow, Glasgow, United Kingdom

*Correspondence to*: Mirjam Schaller (mirjam.schaller@glasgow.ac.uk)

**Abstract.** Surface denudation rates, a composite of physical erosion and chemical weathering, are governed by the tectonic, lithologic, climatic, and biotic conditions of a landscape as well as anthropogenic disturbances. Quantifying rates and disentangling their causes is challenging but important for understanding and predicting landscape evolution over space and time. In this study, we focus on a low-relief and mixed lithology mountain range (Swabian Alb, Southwest Germany), whose 200 to 400 m high escarpment and foreland drain to the Neckar River to the north and whose plateau drains into the Danube River to the southeast. These two drainage systems are subjected to similar uplift rates and climate and biotic conditions but contain different lithologies, base-levels, and topography. We calculate decadal time scale chemical weathering and physical erosion rates based on 43 locations with suspended and dissolved river load measurements and compare them to published longer-term rates (e.g., denudation, incision, uplift).

Chemical weathering rates (based on the dissolved river load, discharge, and corrected for atmospheric and anthropogenic input) range from 0.003 to 0.070 mm/yr, while physical erosion rates (calculated from suspended river load and discharge) range from 0.001 to 0.072 mm/yr. The catchment-wide denudation rates range from 0.005 to 0.119 mm/yr, resulting in ratios of chemical weathering over total denudation rate (*W/D*) between 0.40 and 0.99. These high values indicate that chemical weathering is generally the dominant denudation process in this cool to temperate, humid setting dominated by sedimentary rocks. Both physical erosion and chemical weathering rates are higher in tributaries draining towards the North/Neckar River than in rivers draining towards the Southeast/Danube River, resulting in southeast escarpment retreat rates of 1.0 to 8.1 mm/yr. The anthropogenic effects on denudation rates were evaluated using the Human Footprint and Connectivity Status Indices (HFI, CSI, respectively) as well as the area of artificial constructions for each catchment. After a simplified correction for either index, the natural (non-anthropogenic) denudation rates are estimated to be lower than the values reported above, although it is unclear how to accurately correct rates with either index. Regardless of how correction of anthropogenic impact is applied, we find denudation rates are consistently higher for the Neckar Swabian Alb tributaries.

Comparison of *W/D* values from the Swabian Alb to other study areas in different tectonic, lithologic, and climatic settings with *W/D* values ranging from 0.1 to 1.0 suggests the *W/D* values in the Swabian

Alb (> 0.4) result from high and lithology dependent chemical weathering rates. The high $W/D$ ratio likely results from late-Cenozoic base-level lowering of the Neckar River that resulted in south to southeast-directed escarpment retreat across Southwest Germany. Differences in chemical weathering and physical erosion rates across the escarpment divide may arise from either the contrast in topographic relief, or exposure of lithologies in the Neckar catchment that are more susceptible to chemical weathering and physical erosion.

## 1 Introduction

Landscape denudation rates are influenced by tectonics, lithology, climate, biota, and anthropogenic land use. Denudation is the composite of physical erosion and chemical weathering by biotic and abiotic processes (e.g., Dietrich and Perron, 2006; Schaller and Ehlers, 2022). Disentangling and quantifying spatial and temporal variations in weathering and erosion from biotic and abiotic processes is challenging due to poorly understood interactions between these processes and limitations in the measurement techniques used. However, rates integrating over different time scales can be used to address a broad suite of connections among denudation, short-term anthropogenic impact and land use, and long-term geological processes including active tectonics and ecosystem dynamics (e.g., Hewawasam et al., 2003; Vanacker et al., 2007; Kirby and Whipple, 2012; Sharma and Ehlers, 2023; Ehlers et al., 2022). They can also be used to improve our understanding of the geologic $CO_2$ budget and its influence on global climate (e.g., Raymo et al., 1988; Maher and Chamberlain, 2014; Bufe et al., 2024). Here we evaluate the physical erosion and chemical weathering rates for two large catchments (the Neckar and Danube Rivers) in Southwest Germany that have different base-levels and form a continental divide across the Swabian Alb escarpment. In doing this, we investigate how differences in lithology, topography, and base-level between these catchments contribute to differences in chemical weathering and physical erosion rates.

Previous studies have laid the groundwork for understanding how the mass balance of landscapes and different measurements can be used to quantify chemical weathering, physical erosion, and total denudation rates (e.g., Gaillardet et al., 1999; Riebe et al., 2003; von Blanckenburg et al., 2012). In the following, a conceptual overview is presented for the relevant processes (Figure 1A), and governing equations considered in this study (Figure 1B). Measurements available for quantifying the mass balance are sensitive to processes recorded over different time scales. There is currently no accepted procedure to bridge across the different time scales used by methods presented here, but we highlight their individual sensitivities as they are implicit in the approaches presented. For example, over decadal time scales (Figure 1, left side), catchment-wide denudation rates ($D$) are often determined by making use of catchment area ($A$), river discharge ($Q_w$), and total suspended and dissolved solids ($TSS$ and $TDS$, respectively). Measurement of the previous quantities allows determination of the physical erosion rates ($E$) and chemical weathering rates ($W$; e.g., Gaillardet et al., 1999; Meybeck, 1986). The denudation rates based on river load include (amongst other things) deep weathering of bedrock and saprolite but are problematic for capturing bedload transport and sediment transport during infrequent but large-magnitude events (e.g., Turowski et al., 2010). Over millennial time scales (Figure 1, right side), catchment-wide denudation rates ($D$) for quartz-bearing lithologies can be determined with (for

example) in situ-produced cosmogenic $^{10}$Be in river sand (e.g., Brown et al., 1995; Granger et al., 1996). Combining these rates with measurements from immobile elements (e.g., Zr) in soil and
80 unweathered bedrock allows the partitioning into $E$ and $W$ (Riebe et al., 2003). Calculation of the ratio of chemical weathering rates over total denudation rate ($W/D$) provides a simple metric for understanding the relative strengths of chemical vs. physical processes active (e.g., Riebe et al., 2004; West et al., 2005; Ott et al., 2023). For example, $W/D$ values range from 0 to 1 and reflect total denudation governed by physical erosion, $E$ ($W/D = 0$), or chemical weathering, $W$ ($W/D = 1$). Deep
weathering in landscapes over millennial time scales (Figure 1, right side) is quantifiable by measurement of immobile elements collected from Zr concentrations measured in soil, saprolite, and unweathered bedrock (e.g., Dixon et al., 2009; Riebe and Granger, 2013; Regard et al., 2016). However, the time-intensive quantification of weathering rates with immobile elements is often replaced by measuring $TDS$ (anions, cations) and $Q_w$ (Figure 1, left side; e.g., Campbell et al., 2022).

Recent work attempting to understand catchment-wide $D$ have not only been expanded from $^{10}$Be to in situ-produced $^{36}$Cl in carbonates but also to incorporate the effect of deep weathering with $TDS$ (e.g., Ryb et al., 2014). It has also been shown that the coupling of in situ-produced $^{10}$Be in quartz and $^{36}$Cl in carbonates from river sand allows the determination of both $E$ and $W$ (e.g., Ott et al., 2022; Ott et al., 2024). In addition, $E$ and $W$ of basaltic, silicate, and carbonate lithologies are more often determined
with the technique of meteoric $^{10}$Be (e.g., von Blanckenburg et al., 2012; Wittmann et al., 2015; Dannhaus et al., 2018; Wittmann et al., 2024). Such $D$ have also been compared to rates derived from in situ-produced $^{10}$Be in rivers draining silicate lithologies (e.g., VanLandingham et al., 2022).

One of the common challenges, also relevant to this study, is the quantification of the spatial and temporal variability of $W$, $E$, and $D$ recorded by different approaches (e.g., left vs. right side of Figure
1). For example, rates based on river dissolved solids (over decadal time scales) and in situ-produced cosmogenic nuclides (over millennial time scales) encompasses vastly different time scales. Rates derived from in situ-produced cosmogenic $^{10}$Be may integrate processes active from the Last Glacial Maximum to the present, whereas rates based on river load span only the duration of time over which the measurements are recorded (e.g., 10s to ~100 years). In contrast, the method outlined by Riebe et al.
(2003) reports both $W$ and $E$ over thousands of years. However, this method may be limited by the spatial distribution of lithologies (e.g., Burke et al., 2007; Heimsath and Burke, 2013) and denudation hotspots (e.g., Larsen et al., 2014). Hence, each method and, more importantly, the combination of different methods to quantify rates are subject to uncertainties as most studies rely on integrating a number of measurements over diverse terrain to infer the overall system behavior.

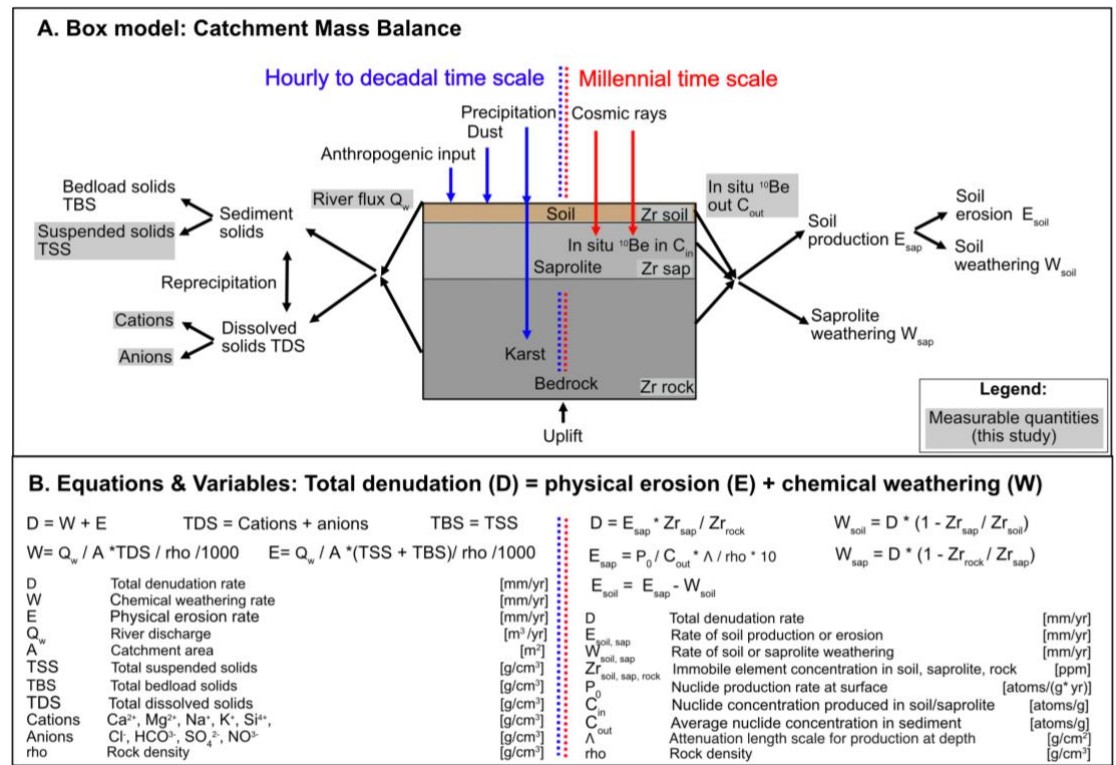

**Figure 1: Approaches for the calculation of denudation rates. Schematic overview of two different approaches to determine chemical weathering and physical erosion and, hence, total denudation rates for river catchments. The method based on river load gauging integrates over hours or the last tens of years (left side). The method of in situ-produced denudation rate in combination with immobile elements integrates over tens of thousands of years (right side). (A) A box model indicating material fluxes and processes over these two time scales, and (B) the equations applied to these time scales, respectively.**

In this study, we complement previous work by investigating decadal time scale $W$ and $E$ using measurements of stream water chemistry, suspended solids, and river flow conditions. We do this with the aim of understanding the partitioning of $W/D$ for adjacent catchments (and sub-catchments) in the same climate zone. More specifically, we investigate intermediate-sized river systems (71 to 12,710 km$^2$) draining the Swabian Alb in Southwest Germany (Figure 2). These rates are compared to millennial time scale estimates of denudation from in situ-produced cosmogenic nuclides and other geologic constraints on escarpment retreat and landscape evolution. Lastly, we also calculate the horizontal retreat rates of escarpments in the study area, as these regions have higher slopes and may differ from catchment-averaged rates. The two rivers lie within a similar climate and ecologic zones, but grade to different base-levels and contain different compositions of layered carbonate, evaporite, and siliciclastic lithologies in their individual sub-catchments (Figure S1). Specific questions we address include: (i) How do lithology and topography determine decadal time scale $W$, $E$, and $D$ in the context of driving escarpment retreat? (ii) How do these decadal time scale rates compare to rates of landscape evolution over millennial time scale and longer time scales? (iii) How do values of $W/D$ in the Swabian Alb compare to global values of $W/D$? And (iv) To what degree are anthropogenic disturbances to the

catchments important for the determination of $W$ and $E$? The results and interpretations presented here suggest that lithologic (e.g., carbonate and sulfate bearing rocks) as well as base-level differences for catchments on either side of the continental drainage divide co-conspire to marked differences in chemical weathering rates. Anthropogenic disturbances in the catchments are significant, but difficult to
robustly quantify. However, after consideration of different approaches to correct for these disturbances, we find the previous interpretations remain the most likely outcome.

## 2 Background to the Swabian Alb

The Swabian Alb is a 200 to 400 m high, 40 to 70 km wide escarpment in Southwest Germany, extending approximately 220 km from southwest to northeast (Figure 2). The escarpment is composed
of Jurassic limestone bedrock that gently dips to the Southeast (0 to 7 degrees), forming a tabular bench (e.g., Dongus, 2000; Thiebes, 2011; Ring and Bolhar, 2020). The Jurassic carbonates are the youngest preserved unit, which caps a 1 to 2-km thick package of alternating sandstone, evaporite, and carbonate stratigraphy (Littke et al., 2008). Triassic and early Jurassic deposits that underly the escarpment encompass a variety of lithologies including siliciclastic formations (e.g., Buntsandstein, Keuper
sandstone, Opalinus clay), carbonates (e.g., Muschelkalk; Arietenkalk, Riffkalk), evaporites (e.g., Keuper gypsum), and pre-Mesozoic crystalline basement rocks (Figure S1). The timing of uplift and subaerial exposure of the Swabian Alb is represented by an unconformity between the Jurassic carbonates and Cenozoic sedimentary cover of the Molasse basin and isolated deposits of Bohnerz (pisolithic iron oxides) in karst fissures with late-Eocene to Pleistocene mammal fossil assemblages
(e.g., Ufrecht, 2008; Ufrecht et al., 2016). In the westernmost area of the Swabian Alb, paleosols of the Bohnerz formation indicate a >5 Ma exposure of this surface based on cosmogenic $^3$He concentrations (Hofmann et al., 2017).

In addition, the Swabian Alb escarpment acts as a significant continental drainage divide (Figure 2). The divide separates rivers on the Swabian Alb that are incised into the Jurassic plateau and flow
southeast into the Danube River, and rivers that start near the escarpment front and flow northwestward through the Jurassic and Triassic foreland towards the Neckar River (subsequently joining the Rhine River; Figure 2). These adjacent river systems exhibit contrasting base-level elevations, with the Neckar River's base-level in Mannheim lying at 83 m above sea level, while the Danube River reaches 465 m in Ulm after a similar along-river distance. The continental drainage divide is characterized by a
pronounced cross-divide topographic asymmetry, implying present-day systematic southward divide mobility due to escarpment retreat (Winterberg and Willett, 2019). A combination of sediment provenance analysis and mapping of fluvial features suggests that the Rhine River and its tributaries have expanded at the expense of the Danube River and its tributaries since the development of the Upper Rhine Graben (Davis, 1899; Petit et al., 1996; Villinger, 1998; Ziegler and Fraefel, 2009). This
transition of formerly Danubian areas to the Rhine River and its tributaries, like the Neckar River, occurred through numerous discrete river capture events, several of which are well-documented (Villinger, 1998; Petit et al., 1996; Ziegler and Fraefel, 2009; Strasser et al., 2010; Yanites et al., 2013).

Previous river load measurements of $Q_w$ and $TSS$ (e.g., Figure 1, left side) suggest decadal time scale erosion rate $E$ spanning from 0.004 to 0.011 mm/yr for both the Neckar River and Danube River

(Blöthe and Hoffmann, 2022; Schaller et al., 2001; DGJ, 2006; DGJ, 2009). Reported weathering rate $W$ in the Neckar River are estimated at approximately 0.020 mm/yr based on $Q_w$ and $TDS$ (Schaller et al., 2001), while denudation rate $D$, the combination of $W$ and $E$, range from 0.023 to 0.027 mm/yr (Schaller et al., 2001). Decadal time scale $W$ from $TDS$ of small tributaries north of the Swabian Alb indicate lithology-specific $W$ (e.g., Hinderer, 2006). The reported rates span from 0.017 mm/yr for

carbonate-rich Keuper sandstone, to 0.038 mm/yr for clay-dominated Middle Jurassic lithologies. $W$ from $TDS$ of the Wutach area, a Danube tributary captured by the Rhine River ~18 kyr ago that cuts through basement, Triassic, and Jurassic rocks, range from 0.045 mm/yr to 0.082 mm/yr with rates exceeding 0.250 mm/yr for evaporite lithologies (Bauer, 1993). Similar high weathering rates for the Wutach area were also reported based on the paired-cosmogenic nuclide method over millennial time

scales (Ott et al., 2024). Springs in the middle Swabian Alb indicate average $W$ of 0.052 mm/yr (Hönle, 1991). Lower rates of 0.027 mm/yr are reported for Upper Jurassic rocks in the Neckar-draining area of Reutlingen/Tübingen (Holzwarth, 1980). These rates are comparable to $W$ from the Aitrach River (0.028 mm/yr), a tributary of the Danube River draining Upper Jurassic rocks (Poppe, 1993).

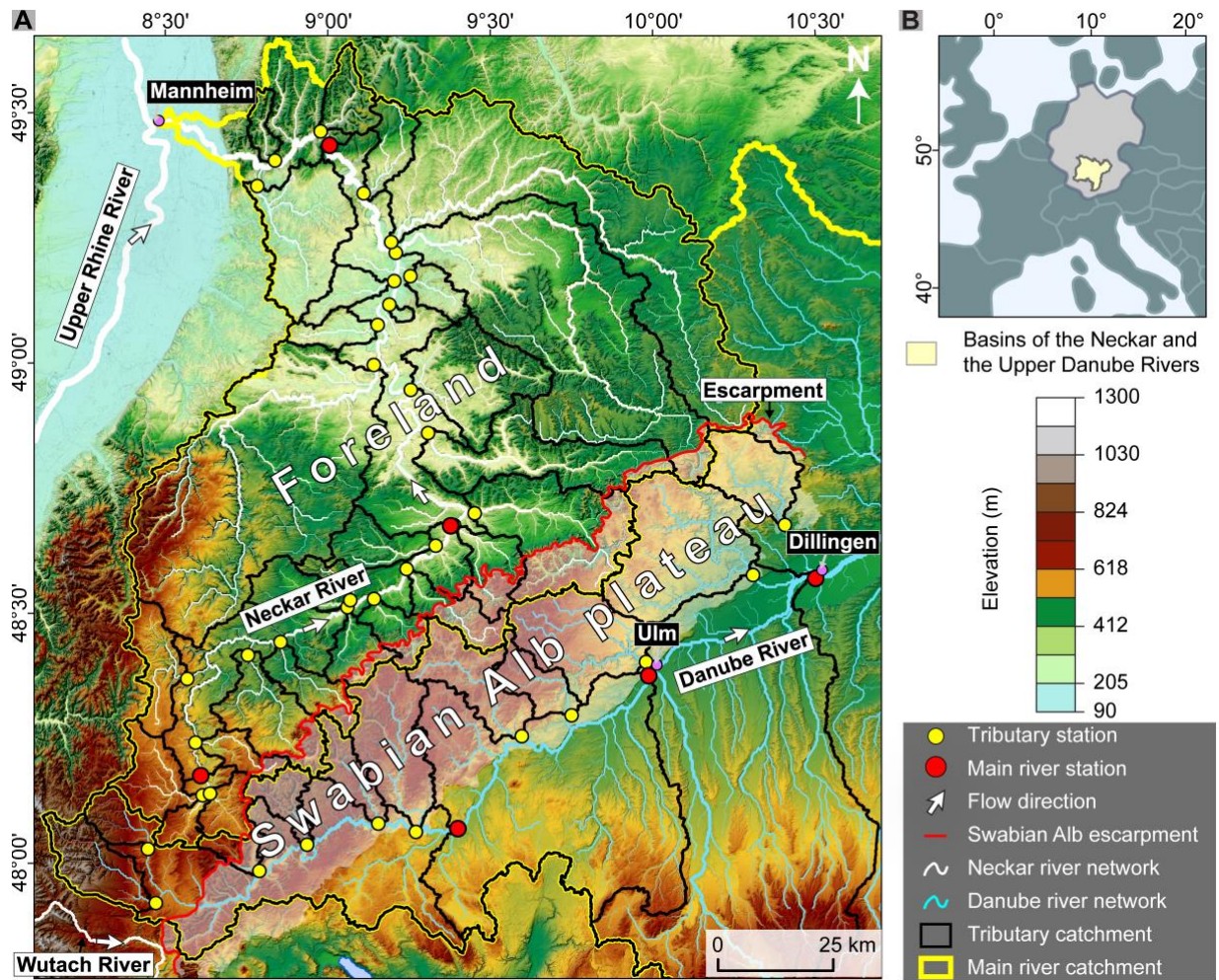

Figure 2: Overview of the study area: Digital Elevation Modell (LGL-BW ATKIS Digitales Geländemodell DGM 5m, 2005) of the Swabian Alb escarpment area (Southwest Germany) with the two main drainage systems of the Neckar River draining Northward the Rhine River in the Northwest and the Danube River draining to the Southeast. The 200-400 m high escarpment is located on the Northwest side of the Swabian Alb plateau and coincides with the continental drainage divide between the Neckar and Danube Rivers. To the Southeast of this drainage divide, the plateau gently dips to the Southeast towards the Danube River. The circles give locations of river measurement stations providing data to calculate physical erosion and chemical weathering rates. See the main text for the range of time periods covered by the observations used.

Previous work suggests that decadal time scale $D$ are generally below the millennial time scale rates derived from cosmogenic nuclide analyses, which range between 0.055 and 0.135 mm/yr (Schaller et al., 2001 and 2002). This has been attributed to a possible under-representation of high-magnitude, low-frequency events in decadal time scale rates or spatially non-uniform denudation in millennial time scale rates due to landslides along the escarpment during the Pleistocene (Terhorst, 2001). In addition, [10]Be-derived denudation rates only reflect erosion of quartz-bearing lithologies, such as Triassic sandstones exposed in the foreland of the Swabian Alb. Long-term rock uplift rates are closer to

existing denudation rate estimates from river loads. Periods of rock uplift above sea level are recorded
by a regionally extensive early Miocene paleo-shoreline (cliff line) preserved along the southern edge of
the Swabian Alb (0.05 mm/yr; Hoffmann, 2017) and by locally preserved cave infills within the
karstified limestone (0.01 mm/yr; Strasser et al., 2009). Cave infills in the Upper Jurassic contain
abundant terrestrial fossils that have been used to create a bio-stratigraphic record of karst evolution
(Ufrecht et al., 2016). Cave levels increase in age, moving to higher elevations within the Swabian Alb,
and fossil assemblages shift to more brackish and marine environments in these oldest deposits,
reflecting the progressive lowering of regional base-level during the late Cenozoic (Abel et al., 2002;
Ufrecht et al., 2016).

## 3 Methods

To evaluate denudation rate $D$ for the Swabian Alb-draining rivers, lithologies and catchment-averaged
metrics are extracted for 3 Neckar River locations, 26 Neckar tributaries, 3 Danube River locations, and
11 Danube tributaries draining the Swabian Alb. Neckar tributaries were separated into tributaries
having a drainage divide with Danube tributaries on the Swabian Alb (12 tributaries called Neckar
Swabian Alb tributary) and all remaining Neckar tributaries in the Swabian Alb foreland (14 Neckar
foreland tributaries). Decadal time scale $W$ and $E$ are calculated from river $Q_w$ and river load $TDS$ and
$TSS$ for three different correction approaches described below. $D$ is then transformed into Swabian Alb
escarpment retreat rates, where applicable. $TSS$ and $TDS$ measurement stations are situated at the same
location. However, the location of $Q_w$ measurement stations may be in some cases at a slightly different
location. The catchment-averaged metrics and the lithologies are extracted at the $TDS$ and $TSS$
locations. The time-periods over which $Q_w$ and river load $TDS$ and $TSS$ were available varied
temporally and spatially due to data availability. In general, $Q_w$ measurements were available from
~1940 to 2009, whereas river load $TDS$ and $TSS$ were available from ~1997 to 2020 (see Table S1 in
data repository for exact date ranges available for each sample location shown in Figure 2).

### 3.1 Catchment-averaged topographic metrics, lithologies, and anthropogenic disturbances

Metrics extracted for the catchments encompass catchment area, mean elevation, local relief, hillslope
angle, local channel slope normalized by upstream drainage area, mean annual precipitation, mean
annual temperature, vegetation cover from NDVI, surface area by lithology as well as anthropogenic
influence. We utilized a 5 m digital elevation model (DEM) sourced from Baden-Württemberg's State
Institute for the Environment (LGL-BW ATKIS Digitales Geländemodell DGM 5m, 2005) to extract
catchment metrics widely used to unravel patterns and rates of physical erosion and chemical
weathering across diverse landscapes (e.g., Ahnert, 1970; Montgomery and Brandon, 2002; DiBiase et
al., 2010; Portenga and Bierman, 2011; Harel et al., 2016). Catchments were extracted from the DEM
with the TopoToolbox software (Schwanghart and Scherler, 2014), employing existing measurement
stations (discharge, solute solids, and dissolved solids) as pour points. Local relief was determined as
the elevation range within a 1.5-kilometer diameter circular neighborhood (e.g., DiBiase et al., 2010;
Peifer et al., 2021). The hillslope angle was computed by fitting a 3-by-3 cell plane for each DEM cell
using the Least Squares Method. A comparison of local channel slope normalized by upstream drainage

area and a regional reference concavity was calculated using the empirical power-law relationship between local channel slope (S) and upstream drainage area (A) (Eq. (1); Flint, 1974):

$$S = k_{sn}A^{-\theta ref} \tag{1}$$

A reference channel concavity ($\theta_{ref}$) of 0.45 was used to compare a normalized fluvial relief ($k_{sn}$) across stream segments or watersheds with different drainage areas (Kirby and Whipple, 2012). The interpretation of $k_{sn}$ is problematic in karstified landscapes such as the Swabian Alb due to substantial subsurface discharge and we urge caution in overinterpreting variations in it. Nevertheless, we include it here for completeness as it is commonly used in similar studies conducted in quartz-rich catchments and

inclusion of it here provides a means for interested readers to compare it to other studies. The river network was extracted using an upslope area threshold for channel initiation of 1 km$^2$ and a smoothing window of 100 m was used to calculate local channel slope. We computed catchment-averaged $k_{sn}$ to compare fluvial relief between basins with suspended sediment and solute load data (Forte and Whipple, 2019).

Catchment-averaged mean annual precipitation rates and mean annual temperature were extracted from the CHELSA (Climatologies at high resolution for the Earth's land surface areas) version 2 climatology dataset covering the years 1981 to 2010 (Karger et al., 2017; Karger et al., 2021). This dataset provides high-resolution (30 arcsec) estimates of precipitation and temperature derived from downscaled ERA-Interim climatic reanalysis. Vegetation cover was estimated using Copernicus Land Monitoring

Service's (CLMS) long-term statistics of NDVI (Normalized Difference Vegetation Index) over the period between 1999-2019 (European Commission Directorate-General Joint Research Centre, 2021; León-Tavare et al., 2021). NDVI, defined as $NDVI = (NIR - Red)/(NIR + Red)$, relies on reflectance measurements in the near-infrared (NIR) and red (Red) bands. Soil thickness is provided by the worldwide data source DAAC (Pelletier et al., 2016).

Lithological data were extracted from the 'General Geological Map of the Federal Republic of Germany' dataset, mapped at a scale of 1:250,000 (BGR, 2019). The extracted lithologies are bundled into 10 bins distinguished by formation age, which are then further classified into three categories, including carbonates (Upper Jurassic), carbonates containing evaporites (Middle Triassic), as well as silicate (Proterozoic, and Paleozoic) together with siliciclastic lithologies (Lower and Upper Triassic, Lower

and Middle Jurassic, Tertiary and Quaternary).

Anthropogenic impacts on each catchment were evaluated using three different main approaches. First, the Connectivity Status Index (CSI) for river systems was evaluated for each catchment, where values of 100% represent undisturbed rivers (Grill et al., 2019). Second, the Human Footprint Index (HFI) was extracted with the highest impact to be 50 and the lowest 0 (Mu et al., 2022). This HFI represents the

relative anthropogenic influence in each terrestrial biome and is represented as a percentage. Third, the % of different landcover such as artificial surfaces and constructions as well as cultivated area, vineyard or tree covers were extracted from the landcover map of Europe 2017 (Malinowski et al., 2020). The third metric used for anthropogenic impact is the % area of artificial constructions and surfaces.

### 3.2 Calculation of decadal time scale rates from river load

$A$, $Q_w$, $TSS$, and $TDS$ of rivers are used to calculate decadal time scale $E$, $W$, and $D$ (Figure 1, left side). $TSS$ and $TDS$ measurement stations generally are situated at the same location (Figure 2). If there was no $Q_w$ measurement station at the same location, the closest $Q_w$ measurement station was selected (24 of total 43 locations; Table S1). The different data sets neither cover the same time periods, not contain the same amount of data, nor was the collected sample material analyzed in the exact same way. Hence, the

calculated rates need to be considered cautiously as the results presented below are based on the average of the time series available. Data for the average of daily $Q_w$ as well as hourly $Q_w$ are consolidated for the Neckar River (3 stations; DGJ, 2009) and some of its tributaries (26 stations; DGJ, 2009; LUBW, 2023) alongside for the Danube River (3 stations; DGJ, 2006) and some of its tributaries (11 stations; DGJ, 2006; LUBW 2023).

The calculation of $W$ is based on the average $Q_W$ and average $TDS$ (Figure 1 left side; single measurement of cations and anions every month for ~20 years) extracted from 29 stations for the Neckar River and tributaries (LUBW, 2023) and 13 for the Danube River and its Swabian Alb tributaries (LUBW, 2023; GKD, 2023). The $TDS$ measurements generally comprise the total concentration of major cations ($Ca^{2+}$, $Mg^{2+}$, $Na^+$, $K^+$ and $Si^{4+}$) along with $Cl^-$, $SO_4^{2-}$, and $HCO_3^-$. $Si^{4+}$

values were unavailable and $HCO_3^-$ concentrations were derived from reported pH and alkalinity (Cole and Prairie, 2014). To calculate $W$, measured $TDS$ can be corrected with different approaches such as atmospheric and anthropogenic inputs, as well as for secondary calcite precipitation. This study uses three approaches to calculate $W$ (Table S2): 1) a simple rate $W_{simple}$ correcred for 60% atmospheric $HCO_3^-$ (Katz et al., 1985; 2) $W_{corr.}$ corrected for atmospheric $HCO_3^-$, rain input (Agster, 1986), and

attribution of all $Cl^-$ to road salt; and 3) $W_{sec.prec.}$ which is based on the assumption that 90% of the originally dissolved $Ca^{2+}$ concentration have reprecipitated (e.g., Erlanger et al., 2021l) in addition to the correction $W_{corr.}$. We note that $W_{sec.prec}$ is a maximum $W$, but we report this rate to show the effect of secondary calcite precipitation on $W$. The $SO_4^{2-}$ concentration is attributed to be entirely the weathering product of gypsum/anhydrite and not due to the oxidation of Fe-sulfides in clay (e.g., Opalinus clay;

Hinderer 2006). $W_{corr.}$ corrects for rain input, assuming that half of the catchment area is cultivated and the other half is forested. This assumption influences the correction of $W$ for rain input only marginally, as rain input contributes generally less than 5% to $TDS$. $W_{corr.}$ provides a minimum rate as NaCl attributed to road salt could also be released by rock salt, which is present in the middle Triassic Muschelkalk formation. In contrast, $W_{sec.prec.}$ is considered a maximum rate, as a 90% fraction of Ca

repreciptation of Ca is likely a maximum estimate (Erlanger et al., 2021) and some Ca is weathered from gypsum in the Muschelkalk and lower Keuper formations.

  $TSS$ measurements (single measurement every month for ~10 to 20 years) are less abundant than $TDS$ data. $TSS$ measurements, used as a proxy for physical erosion rate $E$ over these time intervals, are generally missing measurements during rare high $Q_w$ events and do not include bedload sediment

fluxes. Three different erosion rates $E$ are calculated to bracket these uncertainties using available $TSS$, $A$, $Q_w$ (Figure 1 left side; Data set S1), and a sediment density of 2.7 g/cm$^3$: 1) $E_{simple}$ is based on the average of all reported $TSS$ and the averaged $Q_w$ resulting in a minimum $E$; 2) $E_{RC}$ is estimated from hourly $Q_w$ values and available corresponding $TSS$ using an empirical relation; and 3) a maximum rate $E_{corr.}$ based on further correction of $E_{RC}$ for an addition of bedload (Figure 1 left side). This correction

assumes that the total bedload *TBS* equals *TSS* for sand-bedded rivers as indicated in the compilation of Turowski et al. (2010).

Three different values for *D* are reported based on previous approaches for calculating *W* and *E*. The different *D* values are reported for available measurement stations and include: 1) $D_{simple}$ is the sum of $E_{simple}$ and $W_{simple}$, representing an uncorrected *D*; 2) $D_{corr.}$ is a corrected *D* based on $E_{corr.}$ (the maximum *E)* and $W_{corr.}$ (the minimum *W*). $D_{corr.}$ is considered the most realistic *D* of the three; 3) $D_{sec.prec.}$ is the sum of $E_{corr.}$ and $W_{sec.prec.}$, suggesting a maximum *D*. Based on the different approaches, the contribution of *W* to *D* is reported by dividing *W* by *D*: 1) $W/D_{simple}$ ($W_{simple}$ over $D_{simple}$) resulting in maximum values for *W/D*; 2) $W/D_{corr.}$ ($W_{corr.}$ over $D_{corr.}$) representing a minimum value and is considered to result in the most realistic values of the three; 3) Values of $W/D_{sec.prec.}$ ($W_{sec.prec.}$ divided by $D_{sec.prec.})$ reporting values between $W/D_{simple}$ and $W/D_{corr.}$.

The vertical $D_{corr.}$ which is considered the most realistic *D*, is transformed into a horizontal retreat rate for the Swabian Alb escarpment (e.g., Wang and Willett; 2021). These rates are derived from converting vertical denudation into horizontal retreat and are considered maximum values. More specifically, this was done by taking the denudation rate over the catchment area and applying the mass eroded per unit of time across the escarpment height in that catchment. This approach assumes all catchment denudation occurs along the escarpment and provides an upper bound on the retreat rate. The retreat rate is performed for one retreat direction (Aspect), which is perpendicular to the mean orientation of the shared drainage divide between two adjacent and competing catchments. This was calculated along the entire length of the escarpment within each catchment. The retreat rates were further corrected by taking into account reduced denudation of the plateau area (Wang et al., 2021). Denudation of the plateau area is given by $D_{corr.}$ from the Danube Swabian Alb tributaries sharing catchment divides with the Neckar Swabian Alb tributaries.

In addition, $E_{corr.}$, $W_{corr.}$, and $D_{corr}$ are analyzed for simple linear correlations with geomorphic, climatic/biotic, lithologic, and anthropogenic impact metrics. The reported correlation sets include: 1) All data of the Neckar and Danube Rivers and the tributaries; 2) The Neckar Swabian Alb tributaries; and 3) The Danube Swabian Alb tributaries.

## 4 Results

### 4.1 Decadal time scale rates from river load

$W_{simple}$ from the Neckar River and its tributaries are highly variable and range over two orders of magnitude, from 0.005 to 0.084 mm/yr (Figure 3A; Table S3 to S6). Tributaries draining the escarpment of the Swabian Alb show generally higher chemical weathering rates than the Neckar foreland tributaries that drain older Mesozoic bedrock units. $W_{simple}$ for the Danube River and its Swabian Alb tributaries are more homogenous and generally below 0.040 mm/yr (Figure 3B). The observations made for $W_{simple}$ are the same for $W_{corr.}$ further corrected for rain and road salt input (Figure 4A; Table 1). The average $W_{corr.}$ for the Neckar Swabian Alb tributaries (0.044 mm/yr) is double the average rate for the Danube Swabian Alb tributaries (0.018 mm/yr). Similar trends are also visible for $W_{sec.prec.}$ but with values as high as 0.300 mm/yr for the Swabian Alb tributaries of the Neckar River and 0.160 mm/yr for

the Swabian Alb tributaries of the Danube River. Whereas $W_{sec.prec.}$ is considered a maximum rate, $W_{corr.}$ represents a minimum rate.


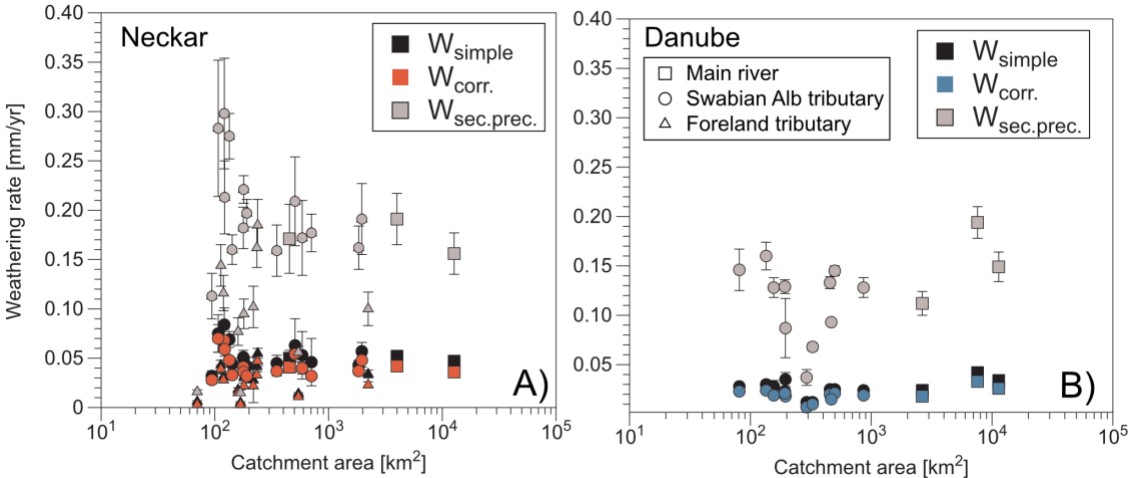

**Figure 3: Decadal time scale chemical weathering rates versus catchment area for: A) the Neckar River (Squares), its foreland tributaries (Triangles), and its Swabian Alb tributaries (Circles) and B) the uppermost reach of the Danube River (Squares) and its Swabian Alb tributaries (Circles). The values shown are the rates corrected for atmospheric HCO₃- ($W_{simple}$), the rates corrected for rain and road salt input ($W_{corr.}$, a minimum scenario), and a rate considering additional secondary calcite precipitation ($W_{sec.prec.}$, a maximum scenario). Values for $W_{corr}$ are considered the most reliable (see methods). Values shown (and in Table 1) are not corrected for anthropogenic disturbances which are addressed in sections 4.2 and 5.1.2.**


$E_{simple}$ based on the average *TSS* and $Q_w$ of rivers ranges from below 0.001 to 0.003 mm/yr for all analyzed rivers (Table S6 and S7). In contrast, $E_{RC}$ can be as high as 0.036 mm/yr for Neckar tributaries but is still low for the Danube River and its Swabian Alb tributaries (0.000 to 0.002 mm/yr; Figure S2).

Correction for bedload increases the $E_{corr.}$ in tributaries of the Neckar River to 0.072 mm/yr, whereas the Danube River and its Swabian Alb tributaries stay below 0.005 and 0.004 mm/yr, respectively (Figure 4B). The resulting $D_{simple}$ range from 0.005 to 0.086 mm/yr, $D_{corr.}$ from 0.005 to 0.119 mm/yr (Figure 4C), and $D_{sec.prec}$ may be as high as 0.346 mm/yr (Table S6). The main general trend is that the rates of Neckar tributaries are more heterogeneous and higher than values from the Danube and its

Swabian Alb tributaries (Figure 4C).

**Table 1: Compilation of chemical weathering, physical erosion, and total denudation rates for the Swabian Alb. (#) Rivers in grey italics are foreland tributaries of the Neckar River. (1) Location and coordinates (WGS84) of measurement stations for dissolved and suspended solids. (2) Chemical weathering rate $W_{corr.}$ corrected for atmospheric HCO₃- and input by rain and road salt. (3)**
**Physical erosion rate $E_{corr.}$ corrected for rating curve and bedload assuming that bedload equals suspended load. (4) $D_{corr.}$ based on $E_{corr.}$ and $W_{corr.}$, (5) $W/D_{corr.}$ based on $W_{corr.}$ over $D_{corr.}$. Values reported are not corrected for anthropogenic impact (see sections 4.2, 5.1.2) aside from rain and road salt corrections, and are based on the procedures described in section 3.2.**

| River | Location [1] | N [1] | E [1] | Area | $W_{corr.}$ [2] | $E_{corr.}$ [3] | $D_{corr.}$ [4] | $W/D_{corr.}$ [5] |
|---|---|---|---|---|---|---|---|---|
| | | ° | ° | km² | mm/yr | mm/yr | mm/yr | |

**Neckar**

| | | | | | | | | |
|---|---|---|---|---|---|---|---|---|
| Neckar | Rottweil | 48.17809 | 8.62060 | 461 | 0.041 | 0.009 | 0.050 | 0.82 |
| Neckar | Wendlingen | 48.67732 | 9.37105 | 3049 | 0.042 | 0.010 | 0.053 | 0.81 |
| Neckar | Rockenau | 49.43178 | 9.00608 | 12647 | 0.036 | | | |
| | | | | **Mean Neckar** | **0.040** | **0.010** | **0.051** | **0.81** |

**Neckar tributaries [#]**

| | | | | | | | | |
|---|---|---|---|---|---|---|---|---|
| *Eschach* | *Rottweil* | *48.13876* | *8.62609* | 217 | *0.022* | *0.001* | *0.023* | *0.94* |
| Prim | Rottweil, Altstadt | 48.14255 | 8.64767 | 121 | 0.068 | 0.008 | 0.076 | 0.89 |
| Schlichem | Epfendorf | 48.24372 | 8.60330 | 110 | 0.070 | | | |
| *Glatt* | *Hopfau* | *48.37126* | *8.57827* | 216 | ***0.033*** | ***0.003*** | ***0.037*** | ***0.91*** |
| Eyach | Mühringen | 48.41900 | 8.75995 | 347 | 0.037 | 0.043 | 0.080 | 0.47 |
| Starzel | Bieringen | 48.44556 | 8.85812 | 178 | 0.041 | 0.030 | 0.071 | 0.58 |
| Steinlach | Tübingen | 48.51550 | 9.05920 | 143 | 0.033 | 0.003 | 0.036 | 0.93 |
| *Ammer* | *Lustnau* | *48.52909* | *9.06789* | 162 | *0.047* | *0.001* | *0.048* | *0.98* |
| Echaz | Kirchentellinsfurt | 48.53152 | 9.13923 | 135 | 0.048 | 0.072 | 0.119 | 0.40 |
| Erms | Neckartenzlingen | 48.59104 | 9.23763 | 179 | 0.036 | 0.004 | 0.040 | 0.91 |
| *Aich* | *Oberensingen* | *48.63758* | *9.32513* | 179 | *0.022* | *0.010* | *0.032* | *0.69* |
| Lauter | Wendlingen | 48.67874 | 9.38075 | 192 | 0.032 | 0.004 | 0.036 | 0.90 |
| Fils | Plochingen | 48.70216 | 9.44380 | 695 | 0.032 | 0.015 | 0.047 | 0.69 |
| Rems | Plüderhausen | 48.86285 | 9.30332 | 575 | 0.040 | 0.010 | 0.051 | 0.79 |
| *Murr* | *Murr-Mündung* | *48.94955* | *9.25137* | *506* | *0.054* | *0.012* | *0.067* | *0.81* |
| *Enz* | *Besigheim* | *48.99929* | *9.13966* | *2214* | *0.023* | *0.022* | *0.045* | *0.51* |
| *Zaber* | *Lauffen, Zaber* | *49.07917* | *9.15174* | *115* | *0.039* | *0.000* | *0.040* | *0.99* |
| *Schozach* | *Heilbronn* | *49.11986* | *9.18743* | *94* | *0.028* | | | |
| *Lein* | *Heilbronn* | *49.16677* | *9.20299* | *115* | *0.028* | | | |
| *Sulm* | *Binswangen* | *49.17688* | *9.25182* | *103* | *0.059* | | | |
| Kocher | Kochendorf | 49.22325 | 9.20766 | 1966 | 0.048 | 0.016 | 0.064 | 0.75 |
| Jagst | Jagstfeld | 49.24538 | 9.19325 | 1834 | 0.037 | 0.005 | 0.042 | 0.89 |
| *Elz* | *Neckarelz* | *49.34209* | *9.10970* | *159* | *0.015* | | | |
| *Itter* | *Eberbach* | *49.46611* | *8.97817* | *155* | *0.003* | *0.001* | *0.005* | *0.73* |
| *Steinach* | *Neckarsteinach* | *49.40799* | *8.83807* | *69* | *0.003* | | | |
| *Elsenz* | *Bammental* | *49.35797* | *8.78512* | *509* | *0.011* | *0.002* | *0.014* | *0.83* |
| | | | **Mean Neckar tributaries** | | **0.035** | **0.013** | **0.049** | **0.78** |
| | | | **Mean Neckar Swabian Alb tributaries** | | **0.044** | **0.019** | **0.060** | **0.74** |

**Danube**

| | | | | | | | | |
|---|---|---|---|---|---|---|---|---|
| Danube | Hundersingen | 48.07238 | 9.39654 | 2629 | 0.018 | 0.004 | 0.022 | 0.82 |
| Danube | Neu-Ulm | 48.37389 | 9.96427 | 7588 | 0.033 | | | |
| Danube | Dillingen | 48.56833 | 10.50016 | 11346 | 0.026 | | | |
| | | | | **Mean Danube** | **0.025** | **0.004** | **0.022** | **0.82** |

**Danube Swabian Alb tributaries**

| | | | | | | | | |
|---|---|---|---|---|---|---|---|---|
| Breg | Hüfingen | 47.92297 | 8.48861 | 269 | 0.007 | 0.003 | 0.010 | 0.70 |
| Brigach | Marbach | 48.03079 | 8.46432 | 135 | 0.018 | 0.002 | 0.020 | 0.88 |
| Elta | Tuttlingen | 47.98806 | 8.79634 | 81 | 0.023 | 0.001 | 0.024 | 0.96 |
| Bära | Hammerschmiede | 48.04055 | 8.93825 | 132 | 0.024 | 0.002 | 0.026 | 0.92 |
| Schmeie | Inzigkofen | 48.08270 | 9.14974 | 153 | 0.019 | 0.002 | 0.021 | 0.91 |
| Lauchert | Sigmaringendorf | 48.06546 | 9.26340 | 454 | 0.020 | 0.002 | 0.022 | 0.91 |
| Grosse Lauter | Lauterach | 48.25587 | 9.58009 | 324 | 0.010 | 0.001 | 0.012 | 0.89 |
| Schmiech | Ehingen | 48.29686 | 9.73059 | 167 | 0.021 | 0.001 | 0.021 | 0.96 |
| Blau | Ulm-Söflingen | 48.40184 | 9.95565 | 486 | 0.021 | | | |
| Brenz | Bergenweiler | 48.57240 | 10.28102 | 755 | 0.019 | 0.001 | 0.020 | 0.95 |

| | | | | | | | | |
|---|---|---|---|---|---|---|---|---|
| Egau | Dischingen | 48.67148 | 10.37964 | 284 | 0.015 | | | |
| | **Mean Danube Swabian Alb tributaries** | | | | **0.018** | **0.002** | **0.020** | **0.90** |

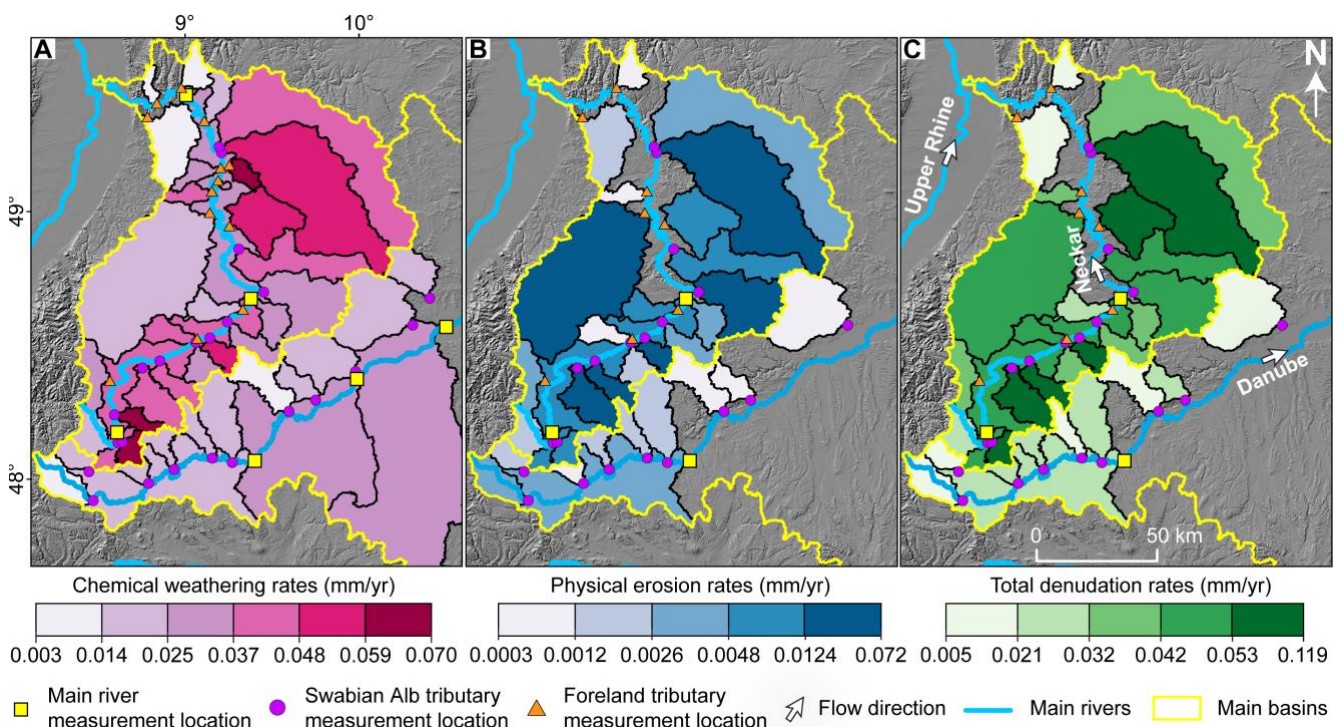

**Figure 4: Mean spatial decadal time scale denudation rates: Maps (LGL-BW ATKIS Digitales Geländemodell DGM 5m, 2005) showing calculated rates in catchments of the Neckar River and Upper Danube River and their tributaries, respectively. A) $W_{corr.}$ derived from dissolved solids. B) $E_{corr.}$ from suspended load and bedload. C) $D_{corr.}$ as the sum of $E_{corr.}$ and $W_{corr.}$. See Table 1 for values plotted.**

Comparing the fraction of chemical weathering on total denudation (Table S6), $W/D_{simple}$ shows little variability (values between 0.94 and 1.00). $W/D_{corr.}$, based on corrections of rain and road salt, yields the strongest spread in values, and $W/D_{sec.prec.}$ shows some outliers for the Neckar tributaries (Figure 5). Generally, $W/D$ values range from 0.40 to almost 1.00 for the Neckar tributaries and from 0.70 to 0.99 for Danube Swabian Alb tributaries. Escarpment retreat rates based on basin projection and $D_{corr.}$ range from 1.0 to 8.1 mm (Average: 3.3 mm/yr; Table 2). Retreat rates which consider lower denudation rates of plateau areas, are only slightly lower (Average: 2.9 mm/yr).

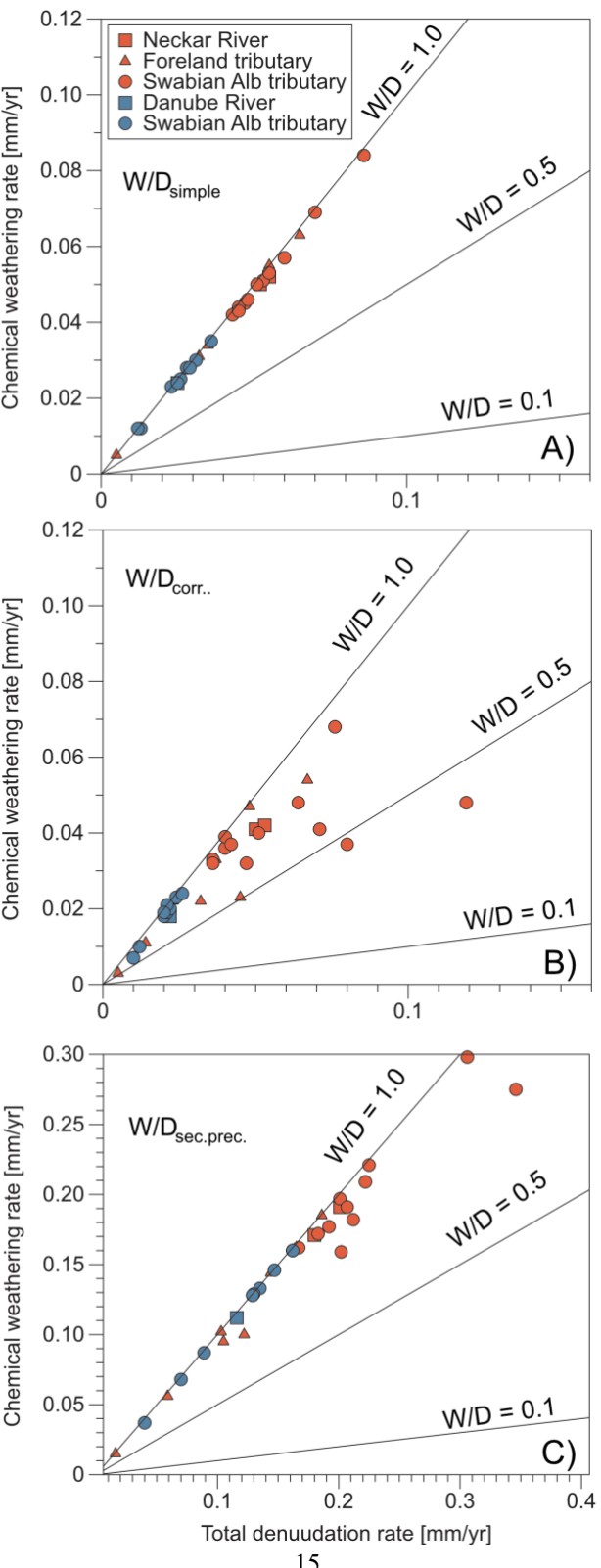

Figure 5: Effect of different corrections applied to the decadal time scale rates: Chemical weathering rate versus total denudation rate for the Neckar River, foreland, and its Swabian Alb tributaries as well as for the Danube River and its Swabian Alb tributaries. A) Uncorrected rates $W_{simple}$ and $D_{simple}$; B) Correction of chemical weathering rates for rain and road salt input ($W_{corr.}$) and $D_{corr.}$; C) Correction of chemical weathering rate for secondary calcite precipitation ($W_{sec.prec.}$) and $D_{sec.prec.}$ See section 3.2 for a description of the different correction approaches.

Table 2: Escarpment retreat rates of the Swabian Alb based on the approach of basin projection provided by Wang and Willett (2021) and Wang et al. (2021). (1) Retreat rate based on $D_{corr.}$ from Swabian Alb Neckar tributaries; (2) Retreat rates corrected for reduced denudation on the Swabian Alb plateau.

| River | | $D_{corr.}$[1] mm/yr | River | | $D_{corr.}$[1] mm/yr | Aspect ° | Basin projection[2] mm/yr | Basin projection[3] mm/yr |
|---|---|---|---|---|---|---|---|---|
| **Neckar tributaries** | | | **Danube tributaries** | | | | | |
| Prim | Rottweil, Altstadt | 0.076 | Elta | Tuttlingen | 0.024 | 135.0 | 2.3 | 2.3 |
| Schlichem | Epfendorf | | Bära | Hammerschmiede | 0.026 | 135.0 | | |
| Eyach | Mühringen | 0.080 | Schmeie | Inzigkofen | 0.021 | 135.0 | 3.9 | 3.8 |
| Starzel | Bieringen | 0.071 | Lauchert | Sigmaringendorf | 0.022 | 135.0 | 2.7 | 2.6 |
| Steinlach | Tübingen | 0.036 | Lauchert | Sigmaringendorf | 0.022 | 157.5 | 1.1 | 1.0 |
| Echaz | Kirchentellinsfurt | 0.119 | Grosse Lauter | Lauterach | 0.012 | 135.0 | 3.6 | 3.3 |
| Erms | Neckartenzlingen | 0.040 | Schmiech | Ehingen | 0.021 | 135.0 | 1.5 | 1.1 |
| Lauter | Wendlingen | 0.036 | Schmiech | Ehingen | 0.021 | 180.0 | 1.3 | 1.0 |
| Fils | Plochingen | 0.047 | Brenz | Bergenweiler | 0.020 | 157.5 | 3.2 | 2.8 |
| Rems | Remsmühle | 0.051 | Brenz | Bergenweiler | 0.020 | 157.5 | 3.3 | 3.3 |
| Kocher | Kochendorf | 0.064 | Brenz | Bergenweiler | 0.020 | 180.0 | 8.1 | 8.1 |
| Jagst | Jagstfeld | 0.042 | | | | 180.0 | 5.8 | |
| **Mean** | | 0.060 | | | 0.021 | 151.9 | 3.3 | 2.9 |

## 4.2 Correlation of catchment metrics with denudation rates

Statistically significant linear correlations with rates ($W_{corr.}$. $E_{corr.}$, and $D_{corr}$ ) for all investigated catchments are reported for only a few metrics (Table 3; Table S8 to S12; Figure S3 to S6). The corresponding P values for all correlations reported here are P <0.05. Furthermore, for brevity, simple linear regressions are not reported for all combinations of rates and metrics and we instead focus on the most meaningful results from the exercise. Results indicate $W_{corr}$ has a moderate inverse correlation with the mean annual precipitation (R = -0.40), some lithologies and anthropogenic impact but no topographic metrics. $W_{corr.}$ shows a moderate positive correlation with anthropogenic disturbances suggested by HFI (R = 0.49) and the area of artificial constructions (R = 0.52) but is not significantly correlated with CSI. Whereas the abundance of the Lower Triassic lithologies (e.g., sandstone) correlates inversely, the Upper Triassic (e.g., sandstone and marl), Lower Jurassic (e.g., claystone), and Middle Jurassic (e.g., claystone) lithologies positively correlate with $W_{corr.}$. $E_{corr.}$ show positive and moderate correlations with maximum relief (R = 0.44), Lower Jurassic (R = 0.46), and Middle Jurassic (R = 0.35). In addition, $E_{corr.}$ correlates inversely with CSI (R = -0.55) and positively with artificial constructions (R = 0.52). $D_{corr.}$ indicate positive moderate correlations with maximum relief (R = 0.41) and a negative correlation with mean annual precipitation (R = -0.40). $D_{corr.}$ correlates negatively with

Lower Triassic (e.g., sandstone) and positively with Lower and Middle Jurassic (e.g., claystone). Furthermore, correlation of $D_{corr.}$ is observed with CSI (R = -0.52), HFI (R = 0.52), and artificial constructions (R = 0.66).

$W_{corr.}$ of the Neckar Swabian Alb tributaries (Table 3 and S11) correlate strongly and inversely with the maximum relief (R = -0.83). In addition, $W_{corr.}$ correlates positively with the percentage of the lithology for the Lower Jurassic (e.g., claystone). A significant negative correlation was found for $E_{corr.}$ with CSI (R = -0.63). In the case of the Danube Swabian Alb tributaries (Table 3 and S12), a moderate to strong positive correlation for $W_{corr.}$ and HFI is reported (R = 0.63). Positive strong correlations of $E_{corr.}$ and

mean elevation (R = 0.82), maximum relief (R = 0.82), and mean annual precipitation (R = 0.79) were found. $E_{corr.}$ has a strong inverse correlation with the mean annual temperature (R = -0.87) as well as cultivated areas (R = -0.75). No meaningful correlation of $D_{corr.}$ with any metric was observed for the Danube Swabian Alb tributaries.

**Table 3: Pearson correlations coefficient between corrected rates and mean catchment metrics, given for all data, Neckar and**
**Danube Swabian Alb tributaries.**

| | All data | | | Neckar Swabian tributaries | | | Danube Swabian tributaries | | |
|---|---|---|---|---|---|---|---|---|---|
| | $W_{corr.}$ n=43 | $E_{corr.}$ n=32 | $D_{corr.}$ n=32 | $W_{corr.}$ n=12 | $E_{corr.}$ n=11 | $D_{corr.}$ n=11 | $W_{corr.}$ n=11 | $E_{corr.}$ n=9 | $D_{corr.}$ n=9 |
| Catchment area | 0.03 | 0.05 | 0.07 | -0.15 | -0.22 | -0.19 | -0.11 | -0.19 | -0.20 |
| Mean elevation | 0.17 | -0.14 | -0.28 | 0.54 | 0.15 | 0.27 | -0.11 | 0.82 | -0.11 |
| Max. relief | 0.07 | 0.44 | 0.41 | -0.83 | 0.08 | -0.28 | -0.30 | 0.82 | -0.50 |
| Local relief (1000m) | 0.09 | 0.23 | 0.09 | -0.26 | 0.24 | 0.08 | 0.15 | 0.38 | 0.23 |
| Trunk mean_$k_{sn}$ | 0.22 | 0.20 | 0.01 | -0.37 | 0.13 | -0.08 | 0.02 | 0.56 | 0.02 |
| Slope | 0.07 | 0.21 | 0.09 | -0.25 | 0.25 | 0.10 | 0.05 | 0.49 | 0.10 |
| Mean annual precipitation | 0.40 | -0.25 | -0.40 | -0.10 | -0.43 | -0.52 | -0.55 | 0.79 | -0.55 |
| Mean annual temperature | 0.22 | 0.16 | 0.31 | -0.57 | -0.12 | -0.25 | 0.18 | -0.87 | 0.18 |
| NDVI (Vegetation cover) | 0.05 | 0.00 | -0.03 | 0.29 | -0.17 | -0.05 | -0.41 | 0.49 | -0.31 |
| Soil depth | 0.20 | -0.08 | 0.11 | 0.57 | -0.11 | 0.12 | -0.30 | -0.17 | -0.48 |
| Connectivity Status Index | 0.20 | -0.55 | -0.52 | 0.19 | -0.63 | -0.57 | -0.24 | -0.08 | -0.21 |
| Human Footprint Index | 0.49 | 0.26 | 0.52 | -0.19 | 0.22 | 0.27 | 0.63 | -0.43 | 0.60 |
| Artificial constructions | 0.52 | 0.52 | 0.66 | -0.07 | 0.51 | 0.51 | 0.41 | -0.03 | 0.40 |
| Cultivated area/vineyards | 0.22 | -0.07 | 0.11 | -0.32 | -0.17 | -0.26 | 0.12 | -0.75 | 0.01 |
| LowerTriassic | 0.44 | -0.11 | -0.31 | | | | | | |
| MiddleTriassic | 0.12 | -0.11 | 0.01 | 0.04 | -0.12 | -0.09 | | | |
| UpperTriassic | 0.47 | 0.11 | 0.35 | 0.21 | -0.11 | 0.04 | | | |
| LowerJurassic | 0.62 | 0.46 | 0.63 | 0.74 | 0.18 | 0.41 | | | |
| MiddleJurassic | 0.41 | 0.35 | 0.44 | 0.06 | 0.13 | 0.10 | | | |
| UpperJurassic | 0.30 | -0.13 | -0.28 | -0.43 | 0.04 | -0.14 | 0.27 | -0.42 | 0.25 |
| Tertiary | 0.26 | -0.16 | -0.25 | | | | 0.00 | -0.53 | 0.03 |
| Quaternary | 0.11 | 0.09 | 0.15 | -0.60 | 0.00 | -0.19 | 0.55 | -0.22 | 0.55 |

$p > 0.05$

$R^2$ 0 to 0.3, $p < 0.05$

$R^2$ 0.3 to 0.5, $p < 0.05$

$R^2$ 0.5 to 0.7, $p < 0.05$

$R^2$ 0.7 to 1.0, $p < 0.05$

## 5 Discussion

The discussion is organized into three sections exploring (i) the decadal time scale chemical weathering $W$, physical erosion $E$, and total denudation rates $D$ across all datasets, with an emphasis on tributaries of the Neckar River and Danube River within the Swabian Alb and also anthropogenic disturbances, (ii) the total denudation rates $D$ observed in the Swabian Alb with rates documented in other studies spanning various longer time scales, and (iii) a discussion of the contribution of chemical weathering and physical erosion to total denudation rates, leveraging global datasets and employing diverse methodologies to ascertain rates.

### 5.1 Decadal time scale rates versus catchment metrics

#### 5.1.1 Interpretation of calculated rates

In general, the $W_{simple}$ and rain/salt-corrected decadal time scale $W_{corr.}$ calculations from all Neckar tributaries (Figure 3A and 4A; average 0.035 mm/yr) agree well with decadal time scale rates reported from small catchments in the Swabian Alb foreland and the Swabian Alb itself (e.g., Hinderer, 2006). These published rates range from 0.017 mm/yr for carbonate-rich sandstone over to 0.038 mm/yr for carbonate-rich claystone and > 0.250 mm/yr for evaporites. The average of chemical weathering rates from these small catchments are comparable to $W_{corr.}$ for larger catchments in this study which contain a mix of these same lithologies. In contrast, the corrected rates $W_{corr.}$ for the Neckar River are higher than the corrected weathering rates of Schaller et al. (2001), which may result from different data sets (e.g., time span and frequency of measurements). The possible correction for secondary calcite precipitation as well as anthropogenic influences on carbonate weathering (e.g., Zeng et al., 2019) introduce additional uncertainties in $W$.

The $E_{corr.}$ (Figure 4B) for the Neckar River are comparable to published rates of ~0.010 mm/yr (Blöthe and Hoffmann, 2022; DGJ, 2006) and ~0.005 mm/yr (Schaller et al., 2001). In contrast, $E_{corr.}$ for the Danube River and its Swabian Alb tributaries are smaller than the already relatively low value of 0.005 mm/yr for the Danube River (DGJ, 2006; Figure 4B). $E_{corr.}$ assumes that sediment load measurements capture a representative distribution of $Q_w$ during the measurement period, but likely under sample infrequent high-magnitude events that contribute high suspended and bedload sediment fluxes relative to solute loads (Pratt-Sitaula et al., 2007; Turowski et al., 2010). In addition, many $TSS$ values decreased in major German rivers (2 000 to 160 000 km$^2$) over the last ~20 years by up to 50% (Hoffmann et al., 2023). Such a decrease in $TSS$ is generally observed in the northern hemisphere due to dams (Dethier et al., 2022). Whereas intensively cultivated small catchments (< 1 km$^2$) may report $TSS$ values more than 40 times higher than pristine catchments (Vanmaercke et al., 2015), such an increase is not observed for larger catchments (>1000 km$^2$). Despite all these uncertainties, a relative comparison shows that $E$ in the Neckar Swabian Alb tributaries are at least two times higher than rates from the Danube Swabian Alb tributaries (Figure 4B). In general, the total denudation rates ($D_{corr.}$), which correspond to a composite of $W$ and $E$, seem to at least two times higher in the Neckar Swabian Alb tributaries than the Danube Swabian Alb tributaries as well as the Swabian Alb foreland tributaries (Figure 4C).

### 5.1.2 Correction of rates for anthropogenic impact

Our correlation analysis between river load rates and various catchment metrics indicates that anthropogenic impact significantly influences calculated $W$, $E$, and $D$ (Table 3; Figures S3–S6). Linear correlations between rates and anthropogenic impact metrics are as strong as, or even stronger than, those observed with lithological and geomorphic variables. A comparison between the Neckar and Danube Swabian Alb tributaries using three anthropogenic netrics — the Connectivity Status Index (CSI), the Human Footprint Index (HFI), and the percentage of artificial surfaces — highlights more intense human impact in the Neckar than the Danube Swabian Alb tributaries (Table S13; see also Methods 3.1). For example, CSI values are on average lower on the Neckar side (99.16%) than on the Danube side (99.54%), indicating more disrupted river connectivity in the Neckar Swabian Alb tributaries. Similarly, HFI values are significantly higher on the Neckar side (29.59%) than on the Danube side (25.19%), and artificial surface coverage is also more extensive on the Neckar side (9.7%) compared to the Danube side (5.2%). All differences are statistically significant ($p < 0.05$), confirming a consistently stronger human footprint in the Neckar than the Danube Swabian Alb tributaries.

To account for these differences, we propose recalculating river load rates by weighting them according to the level of anthropogenic impact. Specifically, $W_{corr.}$, $E_{corr.}$, and $D_{corr.}$ values were adjusted using CSI, HFI, and the percentage of artificial surface area (Table S13). The following weighting formulas were applied:

- CSI-weighted rate = Rate(Table 1) $\times$ ($CSI_{mean}/100$); where 100 represents a fully undisturbed river system.

- HFI-weighted rate = Rate(Table 1) $\times$ (1 - ($HFI_{mean} / 50$)); where 50 is the maximum possible HFI, and higher values denote more human impact.

- Artificial weighted rate = Rate(Table 1) $\times$ (1 - ($Artificial_{mean} / 100$)); where 100 corresponds to complete artificial land cover.

These corrections are based on the assumption that anthropogenic impact such as urbanization and mining typically enhance both physical erosion and chemical weathering by removing vegetation, increasing surface runoff, and promoting contact with reactive agents (e.g., acid mine drainage, polluted urban runoff), among other effects. As such, catchments with higher anthropogenic impact will yield lower adjusted rates than their uncorrected values. Given that the Neckar Swabian Alb tributaries show consistently higher anthropogenic impact, the corrected rates for Neckar Swabian Alb tributaries are generally lower than the uncorrected ones, especially when compared to the Danube side. Among the three correction factors considered, the most substantial adjustments occur with the HFI correction ($D_{HFI}/D_{corr.} = 0.71$), whereas the CSI correction yields the smallest change ($D_{CSI}/D_{corr.} = 0.99$) (Table S13). The positive correlation between HFI and artificial constructions with denudation rates, and the negative correlation with CSI, when all data are considered (Table 3), supports the argument that natural rates are lower than the anthropogenically-influenced ones. We acknowledge, however, that not all human impacts increase erosion rates; for instance, dam construction may reduce downstream sediment

transport and thus lower physical erosion, in which case the adjustment for applying a correction to denudation rates should be reversed.


In summary, the proposed corrections are intentionally simplified, non-process-based, and assume a linear relationship between anthropogenic impact and river load rates. While it remains uncertain which correction method best reflects real-world conditions, our approach offers a starting point for incorporating human impact into denudation rate calculations. Importantly, regardless of the correction

method applied, $W$, $E$, and $D$ in the Neckar Swabian Alb tributaries remain at least twice as high as those in the Danube Swabian Alb tributaries (Table S13). This pattern is consistent even in the uncorrected data (Table 1), indicating a robust signal of elevated rates in the Neckar Swabian Alb tributaries despite the uncertainties surrounding anthropogenic corrections.

### 5.1.3 Interpreted drivers of differences in rates and impact on escarpment retreat

We interpret the remaining differences in $W$, $E$, and $D$ between the Neckar Swabian Alb tributaries and Danube Swabian Alb tributaries to reflect the contrast in base-level between the two main catchments. These base-level differences were set by the reorganization of major river systems since the evolution of the Upper Rhine Graben and onset of long-wavelength uplift in the Middle Miocene (e.g. Ring and Bolhar, 2020). We reason that the contrast in base-level can affect $W$, $E$, and $D$ in two ways. A lower

base-level on the Neckar side of the escarpment (1) increases topographic relief to drive $E$ and $W$ of the subsurface (via flow in karst) and (2) exposes layered stratigraphy with a range of susceptibilities to $E$ and $W$. The climate is comparable on the two sides of the Swabian Alb and the nearly flat-lying Mesozoic strata indicate that differential uplift is minor across the spatial scales of the catchments analyzed, so we focus our discussion on differences in lithology and relief across the escarpment and

their controls on $E$ and $W$.

The steep north-facing escarpment is drained by the Neckar Swabian Alb tributaries eroding the Upper Jurassic caprock and underlying easily erodible rock units of the Middle and Lower Jurassic (Figure S1). In contrast, roughly two-thirds of the area of the Danube Swabian Alb tributaries expose Upper Jurassic carbonates, and exposure of Middle and Lower Jurassic rock is confined to incised valleys

(Figure S1). The occurrence of Lower and Middle Jurassic rock units correlates strongly with $W$, $E$, and hence $D$, which is seen in the Neckar Swabian Alb tributaries (Figure S6; Table 3). A strong correlation of $W$ with the two lithologies may be attributed to claystone at the base of the Middle Jurassic (Figure S6). These claystone units are known to be rich in pyrite (e.g., Hinderer, 2006), which enhances chemical weathering by the release of sulfuric acid (Figure S7; e.g., Ross et al., 2018). Additionally,

these clay units swell and disaggregate when wetted, potentially enhancing physical erosion (Thury, 2002). In this view, the steep topography of the escarpment could be coupled with the higher physical and chemical denudation rates of weak bedrock at the base of the escarpment, which facilitates escarpment retreat. Relief across the escarpment and between valleys of the Danube tributaries drives karstification of the Upper Jurassic caprock in both Neckar Swabian Alb tributaries and Danube

Swabian Alb tributaries, which show a dominance of chemical weathering, and a two-fold contrast in mean rates across the escarpment.

The escarpment retreat rates, calculated using river load and the basin projection method, range between 1.0 and 8.1 mm/yr (Table 2). However, some measurement sites, such as those for the Kocher and Jagst Rivers, are problematic. In these two cases, since the escarpment constitutes only a minor part of the large catchment area, transforming river load into horizontal retreat rates results in a maximum rate. The Swabian Alb escarpment's retreat rates are significantly faster than the global average of 0.6 mm/yr (He et al., 2024) but agree reasonably well with rates of 2 to 10 mm/yr compiled from all around the world (e.g., Duszyński et al., 2019). The relatively rapid retreat rates are likely due to the contrasting base-level elevations between the Neckar and Danube Rivers (e.g., Villinger, 1998; Winterberg and Willett, 2019) and exposure of easily physical and chemically erodible rock units below the Upper Jurassic cap rocks (e.g. claystone). This lithological contrast and elevation difference affects the geometry of these neighboring river systems, with the Neckar River basin growing as the Danube River basin shrinks.

## 5.2 Evaluation of spatial and temporal variations in rates

Despite integrating over short time scales, the higher decadal time scale rates from the Neckar River and its Swabian Alb tributaries than the Danube River and its Swabian Alb tributaries, as well as the escarpment retreat, are reflected in longer time scale surface change rates (Figure 6, Table S14A and 14B). The decadal time scale total denudation rate ($D_{corr.}$) for the Danube Swabian Alb tributaries (Figure 6, blue left-pointing triangles) agrees with millennial time scale rates based on in situ-produced cosmogenic [10]Be in the uppermost reaches of the Danube River (Morel et al., 2003; blue diamonds). Comparable millennial time scale rates from a Danube Swabian Alb tributary are calculated from cave and terrace incisions (Abel et al., 2002; blue dots Figure 6). In addition, the decadal time scale rates agree with uplift rates integrating over the last 17.5 million years (Hoffmann, 2017; blue squares Figure 6). However, the rates determined from cave and terrace incision (e.g., Abel et al., 2002; blue dots; Strasser et al., 2009; red dots; Ufrecht, 2022; downward-pointing red triangles) show local pulses of incision that can exceed rates averaged over million-year time scales or spatial scales of >10 km$^2$ watersheds (Figure 6). For example, an increase in denudation rates is seen at around 3 million years as the Neckar River and tributaries changed their drainage system from the Danube River to the Rhine River. Four pulses of incision are reported from the Laierhöhle cave draining today into the Fils River, a tributary of the Neckar River (Strasser et al., 2009; red dots). Incision pulses 1 to 3 are attributed to the Danube River and indicate rates of 0.004 to 0.017 mm/yr, while pulse 4 is attributed to the Neckar River after drainage reorganization and indicates a rate of 0.700 mm/yr over 0.3 million years in the late Pleistocene. Additionally, an increase in incision rate is reported from terraces of the Kocher River, where rates range from 0.042 to 0.157 mm/yr before 0.78 million years, followed by rates of 0.170 to 0.205 mm/yr at ~0.5 million years (Strasser et al., 2010; downward-pointing blue triangles Figure 6). This transient increase in rates is caused by the capture of the Ur-Kocher (original Kocher River that drained into the Danube River) by the Neckar River (Strasser et al., 2010). These transient pulses of incision are integrated, or average out, by catchment-averaged denudation rates derived from in situ-

produced [10]Be concentrations of river sand, which integrate over millennial time scales and typically exceed decadal time scale estimates of denudation from river load (Figure S8).


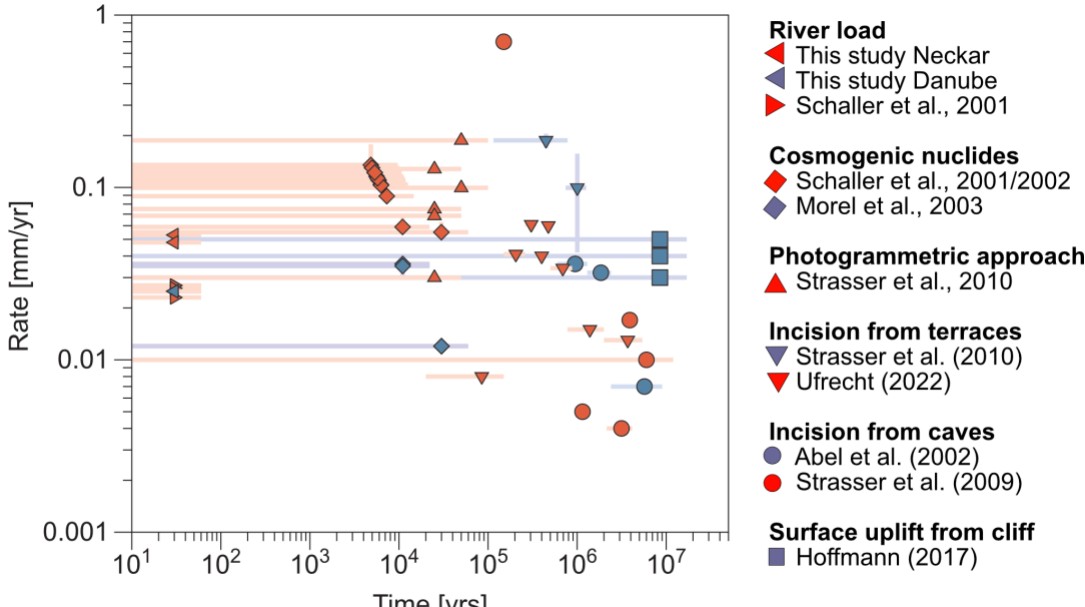

**Figure 6: Synopsis of temporal variations in denudation rates: Compilation of rates integrating over different time scales for locations North (Red) and South (Blue) of the present-day drainage divide of the Swabian Alb. The list of rates ranges from decadal time scale rates from river load to million-years-scale rates based on the uplift of the cliff line in the Swabian Alb.**

To summarize, decadal and millennial time scale rates are more heterogenous and higher on the Neckar-
draining side than the Danube-draining side of the Swabian Alb. In contrast, million-year-scale rates on both sides are comparable and in the range of the decadal time scale rates of the Danube Swabian Alb tributaries. The rates over space and time reflect a consistent picture with lower denudation rates from the plateau and increased denudation rates at the escarpment and its foreland due to continuous escarpment retreat and associated river captures.

**5.3 Evaluation of relative strengths of chemical weathering vs. physical erosion from W/D**

Calculated *W/D* across the Neckar and Danube Rivers are higher than 0.40, indicating that denudation is generally dominated by chemical weathering (Figure 7A). All *W/D*$_{corr.}$ values for the Neckar and Danube Swabian Alb tributaries are higher than 0.40, with averages of 0.74 and 0.90, respectively. This difference is due to higher physical erosion rates in the Neckar Swabian Alb tributaries than in the
Danube Swabian Alb tributaries due to escarpment retreat and river capture events (Figure 4B). The high *W/D*$_{corr.}$ values, which indicate that chemical weathering is the dominant lowering process, are comparable to other German rivers, such as the Weser River (0.99) and the Elbe River (0.95), as well as to the Seine River (0.92) in France (Figure 7A, Table S15A; Gaillardet et al., 1999). The high values are common for catchments in low-relief mountain ranges with mainly mixed sedimentary lithologies under

temperate climatic conditions. However, they are in contrast to *W/D* values of almost 0 from catchments in quartz-rich lithologies (Table S15A; West et al., 2005), in more active tectonic settings (e.g., Amazon (0.19) or Brahmaputra (0.09); Gaillardet et al., 1999), or under different climatic conditions (e.g., Nile (0.27) or Niger (0.18); Gaillardet et al., 1999).

A large spread in *W/D*s is also reported from values based on cosmogenic nuclides for the total
denudation and river load for chemical weathering rates (Figure 7B; Table S15B). *W/D* values from the Apennines, a tectonically active mountain range with mixed lithologies, range from 0 to 1.0 (Erlanger et al. 2021). *W/D* values reported from tropical Cuba with mixed lithologies and tectonic uplift around 0.02 to 0.11 mm/yr (Muhs et al., 2017), range from 0.3 for igneous rocks to 0.96 for sedimentary rocks, respectively (Campbell et al., 2022). In contrast, *W/D* values from Panama, an active tectonic setting
with igneous rocks under a tropical climate, reveal values below 0.4 (Gonzales et al., 2016). Higher *W/D* values than in Panama are reported for metamorphic crystalline basement rocks in tropical climates such as Sri Lanka (von Blanckenburg et al., 2004) and Cameroon (Regard et al., 2016).

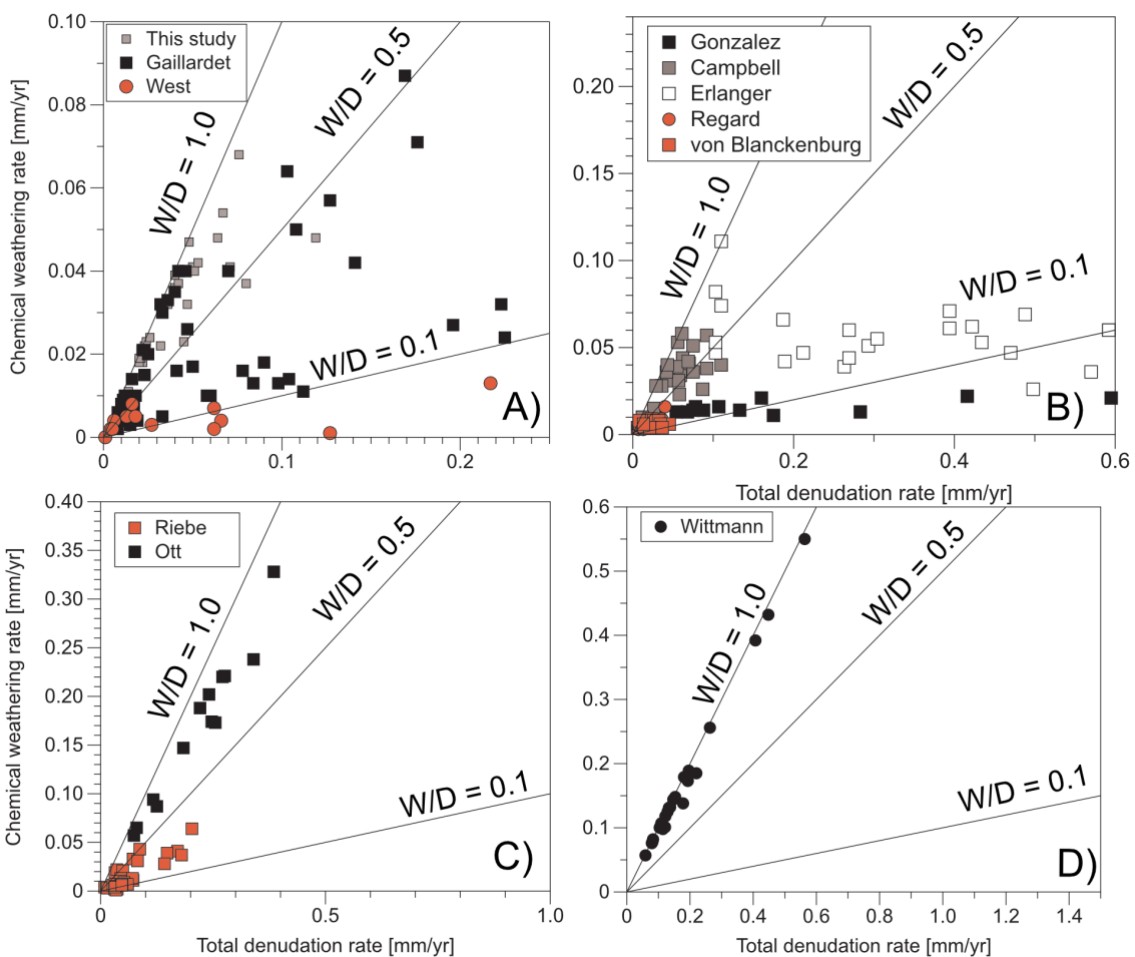

**Figure 7: Comparison of chemical weathering to total denudation ratios (*W/D*) between this study and previous work. Chemical weathering rates versus total denudation rates with lines for *W/D* of 0.1, 0.5, and 1.0. Red symbols indicate data based on catchments**

with dominantly quartz-bearing lithologies, respectively. **A) Data based on river load (e.g., Gaillardet et al., 1999; West et al., 2005; and this study); B) Data based on cosmogenic nuclides and river load in mixed lithologies (e.g., Gonzalez et al., 2016: Erlanger et al., 2021; Campbell et al., 2022) and quartz-rich lithologies (e.g., von Blanckenburg et al., 2004; Regard et al., 2016). C) Data based on cosmogenic nuclides and immobile elements in quartz-rich lithologies (Riebe et al., 2003) as well as paired cosmogenic nuclides in carbonate lithologies (Ott et al., 2024). D) Data based on meteoric $^{10}$Be/$^{9}$Be in limestone (Wittmann et al., 2024).**


Compared to other techniques that integrate over longer time scales and are more representative of landscape evolution (e.g., Riebe et al., 2003; Riebe and Granger, 2013), catchment river loads integrate chemical weathering and physical erosion across diverse bedrock lithologies and include weathering processes that occur in a deeply karstified environment (e.g., Campbell et al., 2022). However, $W/D$
values derived from river loads can be elevated if $E$ derived from suspended sediment flux are underestimated relative to the total sediment fluxes that integrate longer time scales (Pratt-Sitaula et al., 2007). The Neckar and Danube drainage systems both include variations in bedrock stratigraphy and deep weathering processes, and despite the short integration period of river load measurements, calculated denudation rates reflect long-term rates of regional rock uplift and can provide insights into
the mechanics of escarpment retreat in packages of layered sedimentary rock.

Climate, as well as tectonics and lithology, have an important influence on $W$ and $W/D$ values, as reported by Riebe et al. (2004). $W/D$ values (Figure 7C; Table S15C) were calculated based on total denudation from cosmogenic nuclides in river sediment in combination with $W$ derived from the abundance of immobile elements in soil and rock. The method of cosmogenic nuclides in combination
with immobile elements is considered to determine $D$ and $W$ over a similar time scale. Unfortunately, this method of in situ-produced cosmogenic $^{10}$Be in river sediment is restricted to quartz-rich lithologies. The method of in situ-produced cosmogenic $^{36}$Cl can be applied to carbonates for $D$ but still relies on $W$ from river load (e.g., Ott et al., 2019, Ott et al., 2023). The combination of in situ-produced cosmogenic $^{10}$Be and $^{36}$Cl in quartz-bearing lithologies allows the determination of $D$ and $W$ (Figure 7C;
Ott et al., 2024). Another promising new tool to determine $D$ and $W$ over similar time scales in lithologies devoid of quartz (Figure 7D; Table S15D) is the method of meteoric $^{10}$Be in river load (von Blanckenburg, et al., 2012; Wittmann et al., 2015; Dannhaus et al., 2018; Wittmann et al., 2024). Despite the increasing number of methods allowing the quantification of $W$, $E$, and $D$, disentangling the importance of lithology, tectonics, climate/biota, and anthropogenic impact remains challenging. Future
studies (e.g., meteoric $^{10}$Be) should be applied in several simple natural settings differing in only one factor.

### 6 Conclusions

The denudational imprint of tectonics and lithology in a region with similar climate and biota has been addressed using decadal time scale catchment-wide physical erosion and chemical weathering rates
from suspended and dissolved river load in the Swabian Alb (Southwest Germany). Our evaluation of the questions addressed in this study include:

i)  How do lithology and topography determine decadal time scale $W$, $E$, and $D$ in the context of driving escarpment retreat? The results and interpretations presented here suggest that lithologic (e.g.,

carbonate and sulfate bearing rocks) as well as base-level differences for catchments on either side of the Swabian Alb co-conspire to produce marked differences in chemical weathering and physical erosion rates.

ii) How do these decadal time scale rates compare to rates of landscape evolution over millennial and longer time scales? Rates over decadal, millennial, and million-year time scale from the Danube-draining side of the Swabian Alb report relatively homogenous surface change rates close to the uplift rate over the last 17.5 Ma. In contrast, millennial and decadal time scale denudation rates in the Neckar-draining side are generally up to one order higher than in the Danube-draining side, due to catchments recovering from stream capture and escarpment retreat of the Swabian Alb.

iii) How do values of $W/D$ in the Swabian Alb compare to global values of $W/D$? Total denudation rates are generally dominated by chemical weathering, with $W/D$s as high as 0.99. While the Danube Swabian Alb tributaries are governed by chemical weathering, Neckar Swabian Alb tributaries show higher physical erosion due to escarpment retreat and river capture events. Average total denudation rates, and thus morphological activity from the Neckar Swabian Alb tributaries with their higher relief, are two times higher than rates from the Danube Swabian Alb tributaries. Subsequent estimated retreat rates of the Swabian Alb escarpment range from 1.0 to 8.1 mm/yr. Comparable chemical weathering over total denudation rates ($W/D$s) from catchments in different lithologic, tectonic, and climatic/biotic settings reported with river load and in situ-produced cosmogenic nuclides reveal a complex interplay of processes. To better understand rates and processes, several simple natural settings differing in only one factor should be investigated with a single method.

iv) To what degree are anthropogenic disturbances to the catchments important for the determination of $W$ and $E$? Results indicate a strong correlation of decadal time scale rates with different indices for anthropogenic disturbances. The magnitude of these correlations is similar to that as other 'natural' factors considered (e.g., lithology, relief). Despite the significance of anthropogenic processes on rates, after correction of them, the general trend of higher rates in the Neckar rather than the Danube Swabian Alb tributaries persists and meaningful geomorphic interpretations can be made in the region. However, future work is needed to more accurately correct $W$, $E$, and $D$ for anthropogenic impact indices.

**Acknowledgement**

We thank the „Bundesgesellschaft für Endlagerung" for supporting this study by grant STAFuE-21-12-Klei. We would like to thank Richard Ott and Stefanie Tofelde for constructive comments improving the manuscript.

**Data Availability:**

The new data presented in this study used for calculation of chemical weathering, physical erosion, and total denudation are available via https://doi.org/10.5281/zenodo.13588248 (www.zenodo.org).

**Author contribution:**

MS collected the data, wrote text, and made figures with DP. All co-authors wrote specific sections of the manuscript.

**Competing interest:**

No competing interests exist.

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
