# Peer review of "Spatiotemporal denudation rates of the Swabian Alb escarpment (Southwest Germany) dominated by anthropogenic impact, lithology, and base-level lowering"

_EGUsphere, 2024_

## Author Response (AR1)

**General comment to the handling editor(s) and both reviewers.**
We thank Stefanie Tofelde and Richard Ott for their thorough and thoughtful reviews. We appreciate the time and effort they have dedicated to evaluating our work. Nearly all of the suggested changes are very helpful, and we implemented the changes as described in this response to reviews.

General comments were addressed in rewriting section 1 (Introduction) to present a coherent text to the reader. In addition, section 3 (Methodology) has been revised in accordance with the reviewers' suggestions to ensure clarity and robustness. Last but not least, anthropogenic impact was incorporated to the correlation analysis, discussion, and elsewhere in the paper. We note, however, that although we show anthropogenic impacts in the catchments analyzed, there is no straightforward method to quantitatively correct for these effects, so we focus on presenting the relative magnitudes of impacts between catchments. However, as highlighted in the text, regardless of what anthropogenic factors are considered, the general result of this study (high denudation rates in the Neckar vs. Danube catchments) remain consistent.

The specific/detailed reviewer comments are addressed below. The reviewer's comments are in bold, our replies are in italics, and the changed text is in normal font.

**Schaller et al. present decadal rates of physical erosion and chemical weathering for the Swabian Alb and its foreland. The partitioning of denudation into weathering and erosion is calculated analogous to the chemical depletion fraction, and all data are interpreted in the context of climatic, topographic, biologic, and geologic variables. Their findings reveal generally higher and more variable erosion and weathering rates in the Neckar tributaries compared to the Danube, despite notable scatter in the data. The persistence of this general trend is linked to the higher baselevel of the Danube tributaries. The topic of the study is well suited for ESurf. Particularly, the partitioning of denudation in erosion and weathering is of interest—a subject area where more data from carbonate landscapes are essential to better examine the factors that control this ratio. However, anthropogenic influences, which are currently not considered, may significantly affect decadal-scale erosion rates and potentially influence weathering processes. Additionally, a more detailed description of the methods is necessary for clarity and reproducibility.**

*Anthropogenic impact has been addressed by incorporating metrics for anthropogenic influences into the correlation analysis, as well as by expanding the discussion on how physical erosion and chemical weathering rates may be affected by these impacts. The description of the methods is improved, and the basic data (Q, TSS, and TDS) will be accessible. A reproduction of the data should be possible then. As mentioned above, although we can now show anthropogenic impact in the catchments analyzed, there is no clear way to correct it, so we focus on presenting the relative magnitudes of impacts between catchments. However, as highlighted in the text, regardless of what factors of anthropogenic impact are considered, the general result of this study (high denudation rates in the Neckar vs. Danube catchments) remain consistent. Revisions are also reflected in the addition of new Figure S5 and updates to Tables S1, S9, S10, S12, S14, S15, and S16.*

**Human influence: Currently, the study does not take into account human influences on the decadal erosion and weathering rates. The correlations between erosion/weathering and topographic/climatic/geologic variables are weak and a possible explanation would be that the rates are modified by human activity.**

**Parts of the study area are highly industrialized and virtually all of the catchments host significant amounts of agriculture. Moreover, all rivers are heavily modified, channelized, contain hydropower plants of various sizes, canals for mills and factories, flood retention channels, etc. All of these anthropogenic factors should have an effect on erosion and sediment connectivity. For instance, it has been shown that suspended sediment rates in agricultural catchments in Europe can be 40 times higher compared to natural conditions, but this effect is highly scale-dependent due to sediment connectivity (Vanmaercke et al. 2011, 2015). The moderately-sized catchments studied here should be susceptible to these effects. Moreover, sediment connectivity and erosion likely change through time due to changes in land use and river engineering. The study cites very long averaging windows for the data of several decades. It has been shown that river sediment yields are decreasing over the past decades in the Northern Hemisphere, and in Germany in particular (Dethier, Renshaw, and**

Magilligan 2022; Hoffmann et al. 2023). The problem is that the reasons for this decrease are not entirely clear. Since most dams were built earlier, it might rather be related to changes in agricultural practices and/or urban development decreasing hillslope-channel coupling. However, the unknown driver behind these changes make it challenging to select appropriate predictor variables for simple correlations as used here.

All this to say, the human influence on the physical erosion rates needs to be assessed. I encourage the authors to investigate whether variables that capture human activity, such as the human footprint index, % agriculture per catchment, the connectivity status index of rivers (e.g., Grill et al. 2019), etc. better explain the distribution of physical and total erosion rates.

*Different metrics for anthropogenic impact are extracted and incorporated into the analysis for linear correlation. More specifically, for each catchment, we quantified the Human Footprint Index (HFI), Construction Status Index (CSI), and the % area covered by artificial constructions and surfaces. See also the above response for associated text, figures, and tables.*

Similarly, chemical weathering may also be affected by human activity. Soil $CO_2$ is the main source of acid for the dissolution of limestone, and is modulated by vegetation, such that land use differences between catchments may matter in terms of weathering. The land use effects on carbonate weathering have been shown on a global scale (Zeng, Liu, and Kaufmann 2019).

*Some of the different metrics included in the discussion section are the percentage of forest, agricultural land, and populated area. Zeng et al., 2019 is incorporated into the discussion.*

The use of CDF: The chemical depletion fraction has been based on the ratio of element concentrations (mainly [Zr]) in the bedrock and regolith and assuming steady-state soil formation and denudation (Riebe, Kirchner, and Finkel 2003). As such, a CDF represents the long-term contribution of weathering to denudation reflected in the bedrock-regolith composition difference. However, the authors present contemporary rates of erosion and weathering from river sediment and water chemistry. Therefore, I find the use of CDF a bit misleading because the data do not reflect the time-scale of standard CDF measurements. I suggest referring to denudation partitioning, percentage weathering, or similar.

*The reviewer is correct that the use of CDF was incorrect. We changed to using weathering rate over denudation rate named W/D. Text, tables, and figures are corrected for this.*

Just a suggestion, but since the denudation/erosion/weathering rates are low, mm/ka might be the better to unit to report rates and avoid all the zeros.

*The reviewer's suggestion does make sense for the low rates. However, we think that mm/yr makes more sense here as rates are contemporary rates. Therefore, we remain with mm/yr.*

The use of Ksn: The interpretation of Ksn is a bit tricky for this study area, when comparing mostly non-karst (Neckar) to karst-side (Danube) catchments. The relationship between drainage area and discharge is non-trivial in karst landscapes. Therefore, a comparison in Ksn can be difficult or even misleading. One could either use the discharge stations to check for

**drainage area-discharge scaling differences, or at least add a short statement outlining the problems of standard Ksn in karst.**

*Reviewer 1 is correct that Ksn is a tricky measure in karst affected drainage systems. A short statement outlining the problems of standard Ksn in karst was added to this section, for caution. We want to continue to include this analysis in the text as it's standard practice for now karst settings and other readers may want to know it for comparison.*

**More information on the chemical weathering calculation are required. Currently, it is not mentioned how the authors go from ion concentration to a weathering rate.**

- **What discharge data are being used?**
  *The average discharge data $Q_w$ are used for the weathering rate calculations. The average $Q_w$ values are reported in Table S4 and Tables S6B to 6D.*

- **What kind of mineral dissolution is assumed? Probably best to show the equations used for the calculation.**
  *The chemical weathering rate is not based on an assumed mineral dissolution but on measured element concentrations. The three different calculations for the weathering rate corrections are explained in Table S6A and applied to the element concentrations in Table S6B to 6D. We hope these explanations clarify the procedures applied.*

- **How are uncertainties propagated? Fig. 3 shows asymmetric error bars, and almost all data overlap within error. How where these errors computed?**
  *Uncertainties in weathering rates are propagated from the absolute standard deviation ASD of each incorporated element concentration. Figure 3 was misleading for the reader and the three different weathering rates are now displayed in a different way. We hope that the new Fig. 3 is clearer and easier to understand.*

**Significantly more information are required on the calculation of decadal erosion rates. There are a lot of choices to be made to go from a sediment rating curve to a sediment yield estimate:**

- **The authors should report the equation fitted and the a and b values (assuming $Q_s$ = a * $Q_w$^b) as these also hold additional information about sediment transport.**
  *The equation fitted, a and b values, as well as a number of data pairs n and r2, are reported in Table S6.*

- **If the authors fit the rating curve in log-space, what log-transformation bias correction was applied?**
  *The rating curve is fit in a linear space. No transformation correction is needed. This information is added to the manuscript. Text in method section reads now as:" 2) $E_{RC}$ is estimated from hourly $Q_w$values and available corresponding TSS using an empirical power-law relation plotted in linear-space (Table S7);*

- **What was the average number of measurements per rating curve? And what r² values did you get?**

*The number of data pairs n and $r^2$ are reported in Table S7. The number of measurements ranges from 5 to 251. Less than 50 pairs are considered too low for a reasonable rating curve to be calculated. The physical erosion rates with unreliable rating curves are highlighted in TableS7 and mentioned in the method section. In addition, we will add four examples for the rating curves, showing meaningful data and weak data in Fig. S2.*

- **Were quality controls applied, such as minimum amount of data points, minimum $r^2$ value?**
  *See comment above.*

- **Did the authors check for hysteresis in the Qw-Qs data? Frequently, many measurements are taking around flood events, which can bias rating curves due to hysteresis.**
  *The data was not checked for hysteresis mainly due to the low abundance of suspended sediment load data which prohibits a meaningful calculation of it.*

- **The authors cite long time spans for the station records. However, rating curves change over time. Are the entire time periods used for fitting the rating curve? And do rating curves for different river represent different time intervals? Usually rating curves change through time because of human-induced effects on sediment connectivity (Dethier et al. 2022; Warrick 2015) and need to be re-fitted based on a window of interest. For instance, in Germany suspended sediment loads have been decreasing over the past decades (Hoffmann et al. 2023). Therefore, one cannot directly compare erosion rates from rivers with rating curves based on different time intervals or average long-term suspended sediment load. One could decide on a specific decade for analysis or show that temporal trends are negligible.**
  *We agree with the reviewer that the different aspects mentioned are important. However, the data-set for suspended sediment load is too restricted (temporarily) that a specific decade can be picked and compared to another decade as well as another river. Thus, as the reviewer suggests, we've had to restrict our analysis to the time period when the data is available. Given that we are not comparing data over different decades spanning 100 years (for example) the effect mentioned should be minimal.*

- **Most of these questions are also relevant to the other erosion calculation methods, such as the one with average values.**
  *It is true that $E_{simple}$, based on the average suspended load, is suffering from the same problems. Again, the available data-set and time periods available are too small to address such questions.*

**Working my way through the results, I suggest that the authors use more informative names for their rate estimates instead of numbering them. I found it almost impossible to remember what CWR 1,2,3 , PER 1,2,3, TDR 1,2,3, and CDF1,2,3 mean. Can this be simplified to min, max, best guess? Or PER$_{bed}$, PER$_{TDS}$ etc.**

*The numbering is replaced by the following labelling:*
- $W_{simple}$: Uncorrected weathering rate
- $W_{corr.}$: Weathering rate corrected for anthropogenic input
- $W_{sec.prec.}$: Weathering rate corrected for anthropogenic input and secondary precipitation
- $E_{simple}$: Simple erosion from average Qw and suspended load
- $E_{RC}$: Erosion rate based on rating curve
- $E_{corr.}$: Erosion rate from rating curve corrected for bedload
- $D_{simple}$: Sum of $W_{simple}$ and $E_{simple}$
- $D_{corr.}$: Sum of $W_{corr}$ and $E_{corr.}$
- $D_{sec.prec.}$: Sum of $W_{sec.prec}$ and $E_{corr.}$
- $W/D_{simple}$: Division of $W_{simple}$ and $D_{simple}$
- $W/D_{corr.}$: Division $W_{corr}$ and $D_{corr.}$
- $W/D_{sec.prec.}$: Division of $W_{sec.prec.}$ and $D_{sec.prec.}$

We hope the different corrections of W and E  and the resulting D and W/D are clearer and easier to memorize.

**I find CWR3 a bit distractive in the manuscript. I think it's good that the authors acknowledge the fact that secondary calcite precipitation is common in the Swabian Alb and that may lead to an underestimation of weathering if only looking at river chemistry. However, the rates are unreasonably high and the plots with CWR3 and CDF3 could be moved to the supplement. Many studies have calculated chemical weathering rates based on spring water chemistry, such as Hoenle 1991. These data should not be biased by secondary calcite precipitation and the average for the Swabian Alb was 52 mm/ka. Therefore, my recommendation would be to give these maximum estimates less exposure in the manuscript making the paper easier to follow. Also, the authors could compare their rates to the ones previously calculated from spring water chemistry to estimate the amount of secondary calcite precipitation.**

*We agree with the reviewer that the weathering rate corrected for 90% secondary calcite precipitation is a maximum value and may not be realistic for the Swabian Alb area. Nevertheless, we would like to keep the calculations in the main text to show the reader the effect of secondary calcite precipitation.*

*Ideally, we would know the Sr contents in river water in order to do a proper correction for secondary calcite precipitation. Unfortunately, we do not have this information. Following up the suggestion of the reviewer that the weathering rates based on spring water should not be affected by secondary precipitation, we make following thought experiment:*
- *Hönle (1991): average of 0.052 mm/yr for the middle Swabian Alb*
- *Holzwarth (1990): Area Reutlingen/Tübingen: 0.027 mm/yr for Upper Jurassic rocks*
- *Poppe (1993): Aitrach river (Danube tributary): 0.028 mm/yr*
- *This study: Average W of Neckar Swabian Alb: 0.054 mm/yr*
- *This study: Average W of Danube Swabian Alb: 0.025 mm/yr*

*This would mean that there is no secondary calcite precipitation on the Neckar-side or there is an estimated 50% precipitation on the Danube side of the Swabian Alb. However, there is also a low weathering rate from the Neckar-side of the Swabian Alb (Holzwarth, 1990), which would indicate that there is also secondary calcite precipitation there. Hence, it is not clear where in*

*the Swabian Alb and how much secondary precipitation there is in the entire Swabian Alb. A generalized correction of secondary calcite precipitation is not justified, but we cannot address this problem further as there is no data available to do this. We added some of these values to the study area section to bring these values of weathering rates to the reader's attention.*

**Line comments:**

**L10: If we talk about contemporary rates, anthropogenic influences need to be considered.**
*The sentence has been extended by: '…as well as anthropogenic impact in the recent past."*

**L52: No need for granitoid lithologies to measure ¹⁰Be. Please, replace with quartz-bearing lithologies or similar.**
*Granitoid lithologies have been replaced with quartz-bearing lithologies*

**L57-59: I don't follow. A CDF is based on the ratio of two concentrations (bedrock vs saprolith, saprolith vs soil). No need for corrections. Are the authors referring to the need to correct cosmogenic nuclide-derived denudation rates with CDF measurements for weathering below the production zone? Please, clarify or correct this statement.**
*The term CDF has been removed and replaced with W/D standing for the ratio of chemical weathering to total denudation rate. The text is hopefully clearer now.*

**L 69-70: Please, make the time-scales more general. River dissolved loads can integrate over hours; cosmogenic nuclide rates can integrate over hundreds of years depending on the rate. Maybe use powers of ten. For instance, $10^3$ to $10^4$ for cosmogenic nuclides.**
*Text, Fig. 1, and figure caption have been changed to mention that suspended load may include hourly or decadal-scale measurement.*

**Fig 1. Make sure it is clear that the right-hand side is only that complicated in the case of weathering. In an arid environment, cosmogenic nuclide measurements would be tracking denudation.**
*The text has been changed to clarify this:* "However, the time-intensive quantification of weathering rates with immobile elements is often replaced by measuring TDS (anions, cations) and $Q_w$ (Figure 1, left side; e.g., Campbell et al., 2022)."

**L96: Please, add references.**
*Reference is added: Dongus, H., 2000. Die Oberflächenformen Südwestdeutschlands. Gebrüder Borntraeger, Berlin, Stuttgart, 189 pp.*

**L102: Please, add references.**
*Two references are added:*
*Ufrecht, W., 2008. Evaluating landscape development and karstification of the Central Schwabische Alb (Southwest Germany) by fossil record of karst fillings. Zeitschrift Fur Geomorphologie, 52(4), p.417.*

*Ufrecht, W., Bohnert, J. and Jantschke, H., 2016. Ein konzeptionelles Modell der Verkarstungsgeschichte für das Einzugsgebiet des Blautopfs (mittlere Schwäbische Alb). Laichinger Höhlenfreund, 51(21), pp.3-44.*

**L106: It's not just a regional drainage divide. It's part of the continental drainage divide.**
*"Regional" has been replaced by "continental". Thank you for the suggest.*

**L 119-124: Can you please add the methods used for estimating these rates?**
*The measurements used for the determination of rates have been stated briefly. However, we do not detail too much to keep the paragraph short.*

**L121-124: Studies that are missing are for instance the overview work of Hönle 1991 that constrains the average chemical weathering rate of the Swabian Alb to 52 mm/ka, and works from University of Tübingen (!) such as Poppe (1993) and Bauer (1993). Probably, there's more buried in the German literature.**
*Values of some studies mentioned by the reviewer are added to the text in the study area section. The newly included studies are: Hönle (1991), Poppe (1993), Bauer (1993) as well as Ott et al. (2024). However, this list is not exhaustive.*

**L 160: Any justification for 0.45? Many studies use 0.45. However, since the authors are using TopoToolbox, one could use the 'mnoptim' function to check what works best for steady-state channel sections in the area. Though, I understand that heterogeneous lithology might create difficulties here. Nevertheless, please justify the choice of mn.**
*We thank the reviewer for this insightful comment. To estimate how $\vartheta$ varies across the landscape, we calculated the best-fit value of $\vartheta$ for all complete, non-nested tributary catchments of the Neckar River with areas exceeding 10 km² (n = 72). These catchments were extracted using functions implemented in TopoToolbox (Schwanghart and Scherler, 2014). Following the reviewer's suggestion, we used the 'mnoptim' function to determine the optimal $\vartheta$ value for each tributary catchment.*
*Our analysis shows some variation in $\vartheta$ across the Neckar tributary catchments, with a standard deviation of 0.162. However, the mean $\vartheta$ value is 0.431 and the median is 0.480, both of which are very close to the commonly used value of $\vartheta = 0.45$. This provides justification for our choice of $\vartheta = 0.45$ as the best-fit value for the study area.*

**L165-172: Would be good to state how the NDVI time window compares to the measurement time window of erosion and weathering rates.**
*The NDVI time window over 20 years is stated in the method section. This time window is comparable to the window for suspended load (see Tables S3).*

**L173-179: Can you explain on what parameter the geologic binning is based on? Is this based on assumed weatherability, erodibility, or something else? I am confused as to why the basement rocks in the Black Forest end up in the same category as the Keuper evaporites, and Jurassic marls.**

*We tried to clarify how the 10 bins were defined and how the combinations to three bins were performed. The paragraph reads now now: "Lithological data were extracted from the 'General Geological Map of the Federal Republic of Germany' dataset, mapped at a scale of 1:250,000 (BGR, 2019). The extracted lithologies are bundled into 10 bins distinguished by formation age (Table S2), which are then further classified into three categories, including carbonates (Upper Jurassic), carbonates containing evaporites (Middle Triassic), as well as silicate (Proterozoic, and Paleozoic) together with siliciclastic lithologies (Lower and Upper Triassic, Lower and Middle Jurassic, Tertiary and Quaternary)."*

**Table 3: I assume that the categorical variables for geology are converted to % of catchment area? Can you please clarify this in the table or caption, otherwise it is hard to follow what a correlation coefficient of -0.4 between CWR and Lower Triassic is supposed to reflect.**
*Yes, the reviewer is correct and it is % catchment area. We added the units to the metrics and hope that this clarifies the issue.*

**Also Table 3: It just says CWR/PER/TDR. Can you please add the number qualifiers to the column headings? I had to search in the text, which numbers are being displayed.**
*CWR/PER/TDR is now W/E/D and the values used for correlation are the corrected ones ($W_{corr.}/E_{corr.}/D_{corr.}$).Thank you for requesting this clarification.*

**Discussion 5.1.3. Why are these rates only put into global context, and not with regional studies? And what would these rates mean for long-term landscape evolution of the Swabian Alb foreland How do they compare with previous estimates?**
*We discuss the rates with those from regional studies, e.g., those that we have already compiled. In addition, we will state the range of general rates for Germany to highlight the "hotspot" of horizontal retreat.*

**Figure 6: I think this figure would be easier to read if the horizontal bars are removed. The information could be displayed or categorized differently. For some categories different colors refer to geographic location (Danube vs Neckar) for other symbols the color refer to the references. I recommend to revise the layout of this summary figure to make it easier for the readers to follow.**
*We clarified the legend so that the colours refer to the geographical location only. Hence, only red and blue colours are used. The horizontal bars are not removed but are less pronounced. We kept the bars in as they indicate over which time-span the reported rate is valid. We hope that this clarifies the figure.*

**Feel free to contact me in case there are any questions about this review.**
**Richard Ott.**

**References**

Bauer, Michael. 1993. "Wasserhaushalt Und Losungsaustrag Im Wutachgebiet." Pp. 189–202 in *Eintiefungsgeschichte und Stoffaustrag im Wutachgebiet (SW-Deutschland)*, edited by G. Einsele and W. Ricken. Tübingen: Tübinger Geowiss. Arbeiten (TGA).

Dethier, Evan N., Carl E. Renshaw, and Francis J. Magilligan. 2022. "Rapid Changes to Global River Suspended Sediment Flux by Humans." *Science (New York, N.Y.)* 376(6600):1447–52. doi: 10.1126/SCIENCE.ABN7980/SUPPL_FILE/SCIENCE.ABN7980_SM.PDF.

Grill, G., B. Lehner, M. Thieme, B. Geenen, D. Tickner, F. Antonelli, S. Babu, P. Borrelli, L. Cheng, H. Crochetiere, H. Ehalt Macedo, R. Filgueiras, M. Goichot, J. Higgins, Z. Hogan, B. Lip, M. E. McClain, J. Meng, M. Mulligan, C. Nilsson, J. D. Olden, J. J. Opperman, P. Petry, C. Reidy Liermann, L. Sáenz, S. Salinas-Rodríguez, P. Schelle, R. J. P. Schmitt, J. Snider, F. Tan, K. Tockner, P. H. Valdujo, A. van Soesbergen, and C. Zarfl. 2019. "Mapping the World's Free-Flowing Rivers." *Nature* 569(7755):215–21.

Hoffmann, Thomas O., Yannik Baulig, Stefan Vollmer, Jan H. Blöthe, Karl Auerswald, and Peter Fiener. 2023. "Pristine Levels of Suspended Sediment in Large German River Channels during the Anthropocene?" *Earth Surface Dynamics* 11(2):287–303. doi: 10.5194/esurf-11-287-2023.

Poppe, R. 1993. "Karstsystem Und Lösungsaustrag Im Oberen Jura Des Aitrachtals." Pp. 181–88 in *Eintiefungsgeschichte und Stoffaustrag im Wutachgebiet (SW-Deutschland)*, edited by G. Einsele and W. Ricken. Tübingen: Tübinger Geowiss. Arbeiten (TGA).

Riebe, Clifford S., James W. Kirchner, and Robert C. Finkel. 2003. "Long-Term Rates of Chemical Weathering and Physical Erosion from Cosmogenic Nuclides and Geochemical Mass Balance." *Geochimica et Cosmochimica Acta* 67(22):4411–27. doi: 10.1016/S0016-7037(03)00352-X.

Vanmaercke, Matthias, Jean Poesen, Gerard Govers, and Gert Verstraeten. 2015. "Quantifying Human Impacts on Catchment Sediment Yield: A Continental Approach." *Global and Planetary Change* 130:22–36. doi: 10.1016/j.gloplacha.2015.04.001.

Vanmaercke, Matthias, Jean Poesen, Gert Verstraeten, Joris de Vente, and Faruk Ocakoglu. 2011. "Sediment Yield in Europe: Spatial Patterns and Scale Dependency." *Geomorphology* 130(3–4):142–61. doi: 10.1016/j.geomorph.2011.03.010.

Warrick, Jonathan A. 2015. "Trend Analyses with River Sediment Rating Curves." *Hydrological Processes* 29(6):936–49. doi: 10.1002/HYP.10198.

Zeng, Sibo, Zaihua Liu, and Georg Kaufmann. 2019. "Sensitivity of the Global Carbonate Weathering Carbon-Sink Flux to Climate and Land-Use Changes." *Nature Communications* 10(1):1–10. doi: 10.1038/s41467-019-13772-4.

*We thank the reviewer for adding the references to these very interesting publications. We include as many as possible in the study area and discussions sections.*

**General comments**
**In their manuscript 'Spatiotemporal denudation rates of the Swabian Alb escarpment (Southwest Germany) dominated by base-level lowering and lithology' Schaller et al. investigate decadal-averaged chemical weathering and physical erosion rates for a range of river systems draining both sides of the Swabian Albs into the Danube and Neckar rivers. In general, most catchments are dominated by chemical weathering over physical erosion, and total denudation rates (chemical weathering + physical erosion rates) are on average two times higher on the north side of the escarpment draining into the Neckar compared to tributaries draining southwest to the Danube. The authors interpret higher rates in the north to be caused by ongoing escarpment retreat and drainage capture events (caused by base level differences). To better understand spatial variability in erosion and weathering rates, the authors compare rates to topographic, climatic and lithologic catchment parameters and consider lithological differences as an important control parameter on weathering rate differences.**

**While a detailed disentanglement of rates and control factors for denudation processes is useful and timely, and the authors put quite some effort into compiling a large dataset, I have three important comments on the current version of the manuscript. I will mention them briefly here and explain them in more detail below. I suggest that these comments will be addressed before publishing this study in ESurf.**

**First, the study should be streamlined more by matching the focus of the introduction with the rest of the manuscript or adjusting the introduction accordingly. In particular, a more detailed motivation, a clear research question and a discussion of the wider implications of the results would strengthen the relevance of the study.**

*The introduction, the discussion, and the conclusion have been streamlined as suggested, with the revised focus of the manuscript now centered on denudation in southwest Germany. We note, though, that there seems to be a difference of opinion here regarding the structure of the introduction. In our experience, studies that focus solely on a specific geographic region often struggle to generate broad interest, whereas framing research around processes tends to resonate more widely. To balance these perspectives, the introduction retains its emphasis on relevant processes and the current understanding of them while being more closely tailored to the specific application in southwest Germany. The discussion and conclusions clearly tie back to concepts raised in the introduction.*
*We note that despite changes made to investigate anthropogenic impact, the results and interpretations of the paper have not changed significantly from the previous version.*

**Second, the manuscript does not contain sufficient documentation of the methods used. Therefore, reproducibility cannot be guaranteed, nor can it be assessed in all cases how reliable the different approaches are, especially when calculating weathering and erosion rates.**

*The methods section is expanded and calculations are more explicitly treated. In addition, the data-sets used for calculation are deposited in Zenodo together with all the supplementary tables (with a DOI number).*

**Third, the data on water discharge, total suspended solids (TSS) and total dissolved solids (TDS) collected over several decades are susceptible to anthropogenic influences, e.g. agriculture, river dams and discharge controls at sluices. The anthropogenic influences on these time series, and the erosion and weathering calculated from them, are largely neglected.Points 2 and 3 were also addressed by the other reviewer, who gave very constructive and detailed feedback on point 3 in particular. I fully agree with his comments and will therefore focus on points 1 and 2 below.**

*As noted in our response to RC 1, the data presentation and analysis have been extended and improved. Please refer to the responses above for further details. Concerning changes in anthropogenic influence with time, this is also discussed above, but we note here also that different data-sets considered in this study (suspended load, water chemistry) cover different time periods over the last ~70 years and thus do not necessarily overlap (See Table S3). Therefore, the data simply do not exist in sufficient fidelity to do what the reviewer(s) suggest. The majority of data cover the last ~20 years, but even here, a temporal analysis is not possible (with statistical significance) due to the sparsity of observations. In addition, although there are dams on some of these rivers, the duration of time considered in this study for different values (for example) like discharge should provide average flow conditions because dams don't hold water for 30 years – the flow is usually seasonally varied. Thus, our consideration of averages should be smoothing out some of the factors raised here.  It's the best way we could envision conducting the analysis with the available data.*

**Specific comments**
**Structure**
- **The introduction largely focuses on the different methods to disentangle erosion and weathering, but the rest of the paper mainly focuses on the specific case study of the Swabian Albs. After reading the introduction, I expected a more methodology focused paper that tests a new method to differentiate erosion and weathering rates. However, the remaining paper reads more like a case study investigating denudation patterns of the Swabian Albs. I suggest to streamline the paper by making clear from the beginning which direction this study is going. If it is supposed to be a more method- based paper, then the proposed approach should be evaluated in the discussion against other available methods mentioned in the introduction. However, if the general idea is to investigate spatial (and temporal) patterns of weathering, erosion, and denudation in the Swabian Albs, including potential control factors, then I suggest to rewrite the introduction with a clearer focus on such study and also discuss the broader implications of the findings. What can we learn from the Swabian Albs about other sites, will the data help us to improve our process understanding, will the data help us to improve numerical landscape evolution models? The broader impact of the work should also be mentioned at the end of the abstract and in the conclusions.**

*See response above to this reviewer's comment. The introduction has been rewritten (along with other parts of the text). We use the introduction to provide background on the relevant processes evaluated as section 2 of the paper provides the more boring (but needed) site specific background.*

- **Related to the point above, I had difficulties following the introduction about the different available methods on separating erosion and weathering rates (mainly lines 46-67). For example, 'decadal-scale catchment-wide denudation rates (D) can be determined by making use of river discharge (Q) and total suspended and dissolved solids (TSS and TDS, respectively), allowing the determination of physical erosion (E) and chemical weathering rates (W).' But how does this work? Without the extra explanation, it is complicated to follow the individual approaches presented here. Also, the text refers to figure 1, however many parameters mentioned in figure 1 are not explained at all (e.g. Zircon-related parameters, erosion and weathering in the saprolite). In fact, the entire box model shown is not really explained as a model as a whole. On the other hand, not all parameters mentioned in the text show up in the model. For clarification, I suggest to rewrite this part of the introduction (if it stays) and make sure that the text better links to the figure and to point out what the general research aim of this study is (e.g. testing the box model, or finding a different approach for the equations in panel B).**

  *Figure 1 has been edited, and the text referring to in the introduction (and now elsewhere in the paper) has been expanded and clarified.*
  *We have added additional references in the paper for the areas mentioned so that readers are not familiar with these established methods to find more information. We prefer to avoid giving all the equations for each method as they are already published. We have also referred to Fig. 1 more in the text. The caption to Figure 1 has been expanded. The purpose of the figure is to provide an overview of mass balance within the system. These changes along with revisions to the introduction hopefully make it clearer.*

**Methodology**
**Individual steps of the analyses need further explanation, as they are not reproducible in the current state.**
- **Calculation of weathering and erosion rates (lines 187-197): How do you deal with the fact that measurements of discharge, TTS and TDS were taken at different locations when calculating chemical weathering and physical erosion rates? Are these values somehow projected to the point locations giving in the tables?**
  *The reviewer is correct that measurements of $Q_w$, TDS, and TSS are not always at the exact same locations. No projection of the point locations is made. Whereas TDS and TSS are at the same location, $Q_w$ is the closest measurement station if not the same. In order to give an overview a new table is compiled and presented as TableS3. We will further state that we need to be cautious with interpretations.*

- **Water discharge values play a crucial role in the calculations of chemical weathering and physical erosion rates. Discharge values were collected over a few decades (line 191). One crucial information that is missing here is to what degree the discharge in those rivers is anthropogenically controlled (dams, sluices, etc.).**
  *We agree with the reviewer that discharge plays a crucial role in the calculation of chemical weathering rates. However, we do not have the data at hand to show the degree of anthropogenic impact on discharge (e.g., discharge used for irrigation) at different time periods. We could look into changes in discharge over a decadal scale. However, this could also be a climatic effect. We note that we are working with average values over the time series since the different data-sets do not temporally overlap. Given that dams don't retain the same water for 30 years (for example) this approach should provide the best possible approach to interpret chemical weathering rates and other parameters.*

- **Which datasets of discharge, TSS and TDS were used for the calculations? Just saying that the data comes from various online available databases is not sufficient (lines 187-189).**
  *The reference to the online dataset is removed. Instead, all data is provided as data set S1 in a data repository of zenodo (www.zenodo.org) with a doi number.*

- **The three different approaches for calculating decadal-averaged chemical weathering rates (lines 198-214) and physical erosion rates (lines 215-220) are not described with enough detail to ensure reproducibility. Please enhance those descriptions by providing all necessary information.**
  *The descriptions are enhanced and hopefully all necessary information is now available to ensure reproducibility.*

  **Also, it will help to briefly summarize the main assumption behind every approach. For example, the values derived by CWR3 are about 3 times higher compared to those of CWR1 and CWR2, implying that the underlying assumptions of each approach play a major role on the final values. The assumptions will help to get a better feeling for how reliable each of the datasets is.**
  *Text is added to clarify the assumptions made for E and w calculations. For instance:"* $W_{sec.prec}$ is a maximum W and is up to 4 times higher than $W_{corr}$.**"**

  **Another example is PER2, where it says that 'PER 2 is estimated from abundant discharge values using the empirical relation between individual discharge and suspended solids' (lines 217-218). How does the discharge and suspended solid data look like, and what is the equation for the empirical relationship? Further below, it is mentioned that a power-law function is used (line 270), but the whole approach should be explained in the method section and the analysis potentially shown in the supplement.**
  *As mentioned in a response to RC1 we provide four examples of how rating curves can look like in the supplement.*

- **Lines 212-214: Provide a very brief summary of how the method of Campbell et al. (2022) works, such that the reader can follow the approach here without reading the referenced paper.**
  *The correction methods are described in more detail in the text and with Table S6A. In addition, Table S6B, C, and D show the proceeding of the three approaches. The reference to Campbell et al. (2022) is removed as redundant at this location of the manuscript.*

- **Lines 228-230: Please briefly justify why you pick those approaches out of the longer list of options.**
  *The selection of $E_{corr.}$, $W_{corr.}$, and $D_{corr.}$ as the most realistic values, is mentioned briefly and also used for the calculation of escarpment retreat rates.*

- **Lines 232-235: Provide a short summary of how escarpment retreat rates were calculated instead of only referring to previous literature. At least mention the main underlying assumption for the transformation. Also, which locations were chosen for calculating the according catchments? In table 2, only river names are listed. Please add the location used for calculations to ensure reproducibility.**
  *Some sentences are added to explain how retreat rates are calculated. The location names are added to Table 2. Furthermore, there is an additional supplementary table containing locations and number ID's for all the different measurement stations.*

- **Section 4.2: What kind of statistical tests were performed here? In particular, which model was used underlying the tests? Simple linear regressions? If so, does it make sense? Giving that for example denudation rates and mean catchment hillslope gradient or mean ksn are known to correlate non-linear, a simple linear regression is not sufficient. Also, it might help to have different color schemes for positive and negative R2 values in table 3, to clearly distinguish between positive and inverse correlations.**
  *We agree with the reviewer that a linear regression is a simple approach for comparing the different metrics analysed with W, E, and D. In principle, each metric combined with a rate should be investigated in more detail to constrain the different regression. However, a linear regression is a first best guess. We note that for the low denudation rates of the catchment investigated here, the typically low slopes (Table 3) within the catchments, the relationship between slope and denudation is essentially linear (e.g., following results of Montgomery and Brandon 2001 EPSL). In other words – we are investigating the 'flat' part of the non-linear curve they document.*
  *More complicated functions (without the data to constrain them) can introduce issues. For example, if you use a polynomial function you will most definitely get a better fit, but this is because this function has more free parameters. In addition, this fit is not comparable or applicable to other metrics where a linear fit is more reasonable. Therefore, based on the above arguments and previously published work (for the low rates in our study area) a linear fit to the data makes the most sense.*

**Technical corrections**

- **Line 11: Unclear what 'these rates' refer to. Is it about disentangling erosion and weathering rates or the impact of tectonic, lithology, climate and biota on these rates? Please clarify.**
  *We tried to clarify the sentence, which reads now as:* "Quantifying rates and disentangling their causes is challenging but important for understanding and predicting landscape evolution over space and time"

- **Line 17: '…published longer-term rates…' What type of rates? Denudation rates?**
  *The sentence is changed to:*" We calculate decadal-scale chemical weathering and physical erosion rates based on 30 locations with suspended and dissolved river load measurements and compare them to published longer-term rates (e.g., denudation, incision, uplift) to evaluate how the different rates influence landscape evolution."

- **Line 17: '…evaluate how these differences…' What differences, between short and long timescales or between weathering and erosion?**
  *The sentence was corrected as described above.*

- **Line 22: Shouldn't it be 'dominant denudation process' instead of 'erosion'?**
  *Correct, erosion is replaced by denudation. Thank you.*

- **Line 23: What are chemical sedimentary rocks? Please clarify.**
  *Chemical is deleted and the use of the term "sedimentary rocks" is correct .*

- **Line 46: What does the 'these' relate to? Please clarify.**
  *These is deleted. The sentence now reads:*" Disentangling rates of surface changes can be achieved through several methods (e.g., Gaillardet et al., 1999; Riebe et al., 2004; von Blanckenburg et al., 2012)."

- **Figure 2: The figure contains a lot of information using different colours, which are not so easy to see. For example, the Neckar river is hard to recognize. Maybe making the elevation colours more transparent will improve the recognition of the study sites and analysed catchments. In addition, I suggest to add a topographic cross section of the Swabian Albs to the figure, ideally with the underlying lithological units. This will help to reader to follow the description of the study site.**
  *Thank you for your thoughtful feedback on Figure 2. We appreciate your suggestions to improve the clarity and provide additional context for the study area. While we recognise the value of including a geological cross-section, we believe that adding it directly to Figure 2 would significantly complicate the visualisation due to the substantial amount of information already presented in this figure. Adding another layer, such as a prominent line representing the spatial distribution of the cross-section, could obscure key elements like study sites and analysed catchments, ultimately reducing the clarity of the figure. Instead, we have included a geological cross-section in Figure S1 of the*

*supplementary material. This cross-section provides detailed information on the regional geology and highlights the lithological units relevant to the study area, offering additional context without overcrowding Figure 2.*

*To enhance the clarity and readability of Figure 2 itself, we have made several improvements. These include adding the spatial distribution of the Swabian Alb escarpment front, indicating the spatial extent of the Swabian Alb Plateau, and clearly marking the foreland region. We have also refined the use of colours and adjusted the transparency of elevation shading to make the study sites, analysed catchments, and the Neckar River more visible and easily recognisable. Additionally, we have focused on clarifying the description and provided references (Dongus, 2000; Thiebes, 2011; Ring and Bolhar, 2020) for the geologic background of the region if readers are interested.*

- **Line 147: What does 'these' refer to? Which metrics exactly?**
*The sentence has been changed and reads now: "Metrics extracted for the catchments encompass catchment area, mean elevation, local relief, hillslope angle, local channel slope normalized by upstream drainage area, mean annual precipitation, mean annual temperature, vegetation cover from NDVI, and surface area by lithology (Table S1 and S2)."*

- **Lines 168-172: NDVI is mentioned in the method section, but no data is presented later in the study. Consider removing:**
*The results section now includes vegetation cover from NDVI, and a figure (Fig. S5) is included in the supplemental material.*

- **2. I suggest to mention somewhere early in this section the number of stations that are analysed. Some numbers are presented later, but it would be helpful to get a good idea of the dataset early on.**
*The number of stations is mentioned at the start of the methods : "* In order to evaluate D for the Swabian Alb-draining rivers, lithologies and catchment-averaged metrics are extracted for 3 Neckar River locations, 26 Neckar tributaries, 3 Danube River locations, and 11 Danube tributaries draining the Swabian Alb (Table S3). Neckar tributaries were separated into tributaries having a drainage divide with Danube tributaries on the Swabian Alb (12 tributaries called Neckar Swabian Alb tributary) and all remaining Neckar tributaries in the Swabian Alb foreland (14 Neckar foreland tributaries). Decadal-scale W and E are calculated from river $Q_w$ and river load TDS and TSS for three different correction approaches. D is then transformed into Swabian Alb escarpment retreat rates, where applicable. TSS and TDS measurement stations are situated at the same location. However, the location of $Q_w$ measurement stations may be in some cases at a slightly different location. The catchment-average metrics and the lithologies are extracted at the TDS and TSS locations."

- **Line 238: Which part of the study area is considered as the plateau? Could it be indicated in one of the maps?**
*We now indicate the spatial extent of the Swabian Alb plateau in figure 2.*

- **Line 267: What is meant by left-side tributaries of the Neckar river? Those draining the foreland and not the Alb?**
  *The term "left-side tributaries" has been removed. We now distinguish now between Neckar Swabian Alb tributaries that share a drainage divide with Danube tributaries and those referred to as Neckar foreland tributaries..*

- **Line 270, point 6: In line 219 the approach is described differently, please clarify.**
  *Line 219 and 270 are adjusted to not contradict each other. Line 219 reads now:"* This correction assumes that bedload equals suspended load for sand-bedded rivers as indicated in the compilation of Turowski et al., (2010*).". Additionally, Line 270 is changed to:"* $E_{corr.}$ is corrected for bedload assuming that bedload equals suspended load

- **Line 350: Misplaced reference to figure 4, or at least unclear what figure 4 is referenced for.**
  *The misplaced reference is removed. Thank you.*

- **Line 371: Did you mean to refer to figure S4? Figure 4 shows no information on lithology.**
  *Yes, reviewer is correct that reference waw not correct. After revision, reference to Fig. S6 is correct.*

- **Line 372: I don't follow why Figure S5 is reference here, the figure shows no information in Opalinus clay.**
  *The reference has been corrected to Figure S6, which includes information on lithology. In addition, the text in Line 372 reads now:"*A strong correlation of W with the two lithologies may be attributed to claystone at the base of the Middle Jurassic (Figure S6)."

- **Line 381-384: This should be moved to the method section.**
  *Text is moved to the method section.*

- **Line 387-389: Is the mismatch between Swabian Albs retreat rates and global retreat rates due to a special setting of the Alb sir rather a methodological bias, based on the way the catchment area is calculated?**
  *We do not think that there is a methodological bias, as there is no actual mismatch between the Swabian Alb's retreat rates and those observed globally. The Swabian Alb's retreat rates align well with other rates reported worldwide. The text has been revised to state:"*The Swabian Alb escarpment's retreat rates are significantly faster than the global average of 0.6 mm/yr (He et al., 2024) but agree reasonably well with rates of 2 to 10 mm/yr compiled from all around the world (e.g., Duszyński et al., 2019)."

- **Figure 6: Numbering (A,B,C,…) in caption is misleading, as there are no sub-plots. Looks like some error bars are of different colours than the according symbol. (C), what kind of DEM analysis is used to calculate surface lowering rates?**

  *The misleading numbering is removed from the figure caption. The figure caption reads now:"* Figure 6: Synopsis of temporal variations in denudation rates: Compilation of rates integrating over different time scales for locations North (Red) and South (Blue) of the present-day drainage divide of the Swabian Alb. The list of rates ranges from decadal time scale rates from river load to million-years-scale rates based on the uplift of the cliff line in the Swabian Alb."

- **Line 428: '…to being dominated by physical erosion…' This phrasing is misleading as the CDFs barely reach values below 0.5. I suggest rephrasing.**
  *This sentence was not only misleading but wrong. The sentence reads now as:"* Calculated W/D across the Neckar and Danube Rivers are generally higher than 0.5 indicating that denudation is dominated by chemical weathering (Figure 7A)."

- **Lines 434-435: Where are the other reported values from geographically? What can be learn from comparing the values of the Swabian Albs with other sites globally?**

  *Some locations and river names have been added to clarify the location and differences between these different areas. The sentence reads now as:"* However, they are in contrast to W/D values of almost 0 from catchments in other lithologies (Table S18; e.g., West et al., 2005), in more active tectonic settings (e.g., Amazon (0.19) or Brahmaputra (0.09); Gaillardet et al., 1999), or under different climatic conditions (e.g., Nile (0.27) or Niger (0.18); Gaillardet et al., 1999)."

- **Line 475-492: Currently, the conclusion is only a summary of the main finding, but lacks any broader implication of the study's finding. I suggest to add a few sentences about what we can learn from this study beyond erosion and weathering rates in the Swabian Albs to enhance to relevance of the study for a more general community.**
  *Conclusions have been changed to match the introduction and to make it useful for a more general community.*

- **Figure S4: Instead of only reporting the geologic units (e.g. Lower Jurassic), I suggest to also provide additional information on the dominant lithologies of the units.**
  *The dominant rock type of the lithological unit is added to the figures.*

---

## Referee Report (RR1)

In their revised manuscript entitled 'Spatiotemporal denudation rates of the Swabian Alb escarpment (Southwest Germany) dominated by anthropogenic impact, lithology, and base-level lowering', Schaller et al. analyse spatial and temporal patterns of weathering, erosion and denudation rates for the Swabian Alb. The authors have substantially modified the earlier version of the manuscript to take into account the reviewers' comments. Although the current version is a great improvement, I still have some comments, which are listed below.

**Methods**

An important comment from both reviewers related to the incomplete description of the methods, which made it difficult to follow the individual approaches presented in the study and hindered the reproduction of the analyses. In the revised version, the authors go into sufficient detail and readability has been improved. I have 4 comments here.

Firstly, it is still unclear how weathering and erosion rates are ultimately quantified using TSS and TDS. Section 3.2 gives a general explanation of which parameters are used in the calculation, but the equations themselves are not given. I assume that the equations shown in Figure 1 on the left are used. I suggest either referring to Figure 1 here or listing the equations in Section 3.2 and possibly removing them from Figure 1 (see also comment below).

Secondly, I appreciate the fact that the authors have now introduced three new proxies to quantify the anthropogenic influence on the catchment areas (lines 264-277). However, it is not clear from the description how the values were derived and what these parameters actually mean. Were the values calculated by the authors or taken from existing data sets? What does the connectivity status index (CSI) describe, lateral or downstream connectivity or both? And if the human footprint index (HFI) is given in % and the highest values found in the study region are ~50 %, why are 50 % and not 100 % used in the calculation of anthropogenic impacts (Section 5.1.2)?

Thirdly, I appreciate that the authors have tried to take into account the anthropogenic influence on erosion, weathering and denudation rates. However, I am a little concerned about the approach used. Although the authors clearly point out that the approach should be taken with caution as there is no standard procedure, I think the approach presented corrects in the wrong direction. Lines 465-465 state: '[...] many TSS values have declined by up to 50% in large German rivers (2 000 to 160 000 km2) in the last ~20 years (Hoffmann et al., 2023). Such a decrease in TSS is usually observed in the northern hemisphere due to dams (Dethier et al., 2022).' This statement implies that human influence has led to a reduction in TSS and thus weathering rates. Natural/unmodified rates would therefore be higher. However, the weighting of the anthropogenically corrected rates proposed here (Section 5.1.2) leads to a reduction in the natural rates compared to the measured human-influenced rates. For example, the greater the human influence, the lower the ratio of $CSI_{mean}/100$ and the lower the weighted natural rates. In view of the above statement, shouldn't the correction go in the other direction and instead increase the natural rates with increased human influence?

Fourth, I am still concerned about the approach to examining the relationship between erosion, weathering or denudation rates with topographic, climatic or biotic average catchment parameters (lines 342-345). Although I understand why the authors favour linear regressions over polynomial regressions, several previous studies have shown non-linear relationships between these rates and catchment average parameters. Therefore, instead of calculating the Pearson correlation coefficient, which assumes linear relationships, I suggest calculating the Spearman rank correlation coefficient instead. This measures the strength and direction of two variables

(monotonically increasing or decreasing), but does not assume linearity. I think this approach makes far fewer assumptions about the underlying relationships, but still provides a metric similar to the one presented in the study.

**Structure**

The readability of the manuscript has been significantly improved compared to the previous version. And I understand that there are different writing and organising styles for a paper. However, I still find the introduction quite complicated, especially the two paragraphs on how denudation, erosion and weathering rates are quantified on different time scales (lines 62-95, Fig. 1). As mentioned earlier, after reading this introduction, I would expect a study that focuses specifically on bridging the gap between denudation rates at different time scales. Instead, much of the results and discussion focus on explaining the spatial variability of denudation rates and identifying parameters that control these rates. This part of the study is not justified in the introduction. It is also not clear from the introduction why the horizontal retreat rates are calculated. Therefore, I would again suggest that the structure of the introduction be better aligned with the content of the rest of the manuscript.

**Technical corrections**

- Sometimes abbreviations like W or E are not in italics (e.g., lines 172, 279, and more).
- Lines 184-185: "…drainage systems of the Neckar River draining Northward the Rhine River in the Northwest and the Danube River draining to the Southeast." The sentence needs to be grammatically corrected.
- Table 2: What is meant by direction here, the aspect in degrees?
- Lines 411-412: Remove italics.
- Table 3 caption: "Correlations between corrected rates and mean…" What kind of correlations are reported here, what parameters are given in the table? I assume it is the Pearson correlation coefficient?
- Line 473: A 'be' is missing here.
- Line 483: The closing parenthesis after 'Footprint Index' is missing.
- Line 688: Remove single letters in the sentence.

---

## Author Response (AR2)

**General comment to the handling editor(s) and both reviewers.**
We thank Stefanie Tofelde and Richard Ott once more for their reviews of the revised manuscript. The suggested changes are very helpful and we implemented the changes as described in this response to reviews. The specific/detailed reviewer comments are addressed. below. The reviewer's comments are in bold, our replies are in italics, and the changed text is in normal font.

**Public justification (visible to the public if the article is accepted and published):**
**Dear authors,**

**Many thanks for submitting your revised manuscript. The two original reviewers have now submitted a further round of reviews, which range from minor to major revisions. Both reviewers appreciate the attempt of quantifying the impact of anthropogenic disturbance in the catchments, which I echo, and it is interesting to see that these metrics correlate as strongly or stronger with denudation rates than the impact of lithology or geomorphic metrics. I think this study will be of great interest for those attempting to quantify anthropogenic impacts on erosion and is a good first step along that road.**

**However, both reviewers feel that more explanation of the different anthropogenic metrics is needed along with further justification of how denudation rates can be corrected to account for anthropogenic impacts. Both reviewers still feel some methodological detail is lacking on the calculation of weathering and erosion rates from TSS and TDS. Reviewer 2 also points out that the introduction is focused on comparing denudation rates across scales. While this is discussed, a lot of the manuscript is focused on spatial differences in denudation rates which could be better emphasised in the introduction.**

**Overall, I do not feel these add up to major revisions, but all these comments should be carefully addressed. Please respond fully to the comments of both reviewers while preparing your revised manuscript.**

**Best wishes,**
**Fiona Clubb (Associate Editor)**

**Response to reviewer comments RC 1**

**The authors have done a good job at addressing my review and the manuscript is much improved. I am glad to see that a clear relationship between erosion and weathering with anthropogenic factors could be documented. Below I list minor comments that should be addressed before publication.**

**In the initial review, I asked what kind of mineral dissolution was assumed. I tried to follow what the authors did in the excel sheets, but it wasn't obvious to me (an equation in the manuscript would really help). I got the impression that all the cations and anions were summed to get the total dissolved solids, which were multiplied by discharge. But does that include HCO3-? Because if CaCO3 is dissolved, HCO3- is partially derived from the water and the CO2 in the atmosphere, which is why I asked about assumed mineral compositions in the first place. I might be wrong and confused here, but would be nice to clarify.**

*We thank the reviewer for their attention to detail on this and for helping us catch this. We (finally) realized that one important correction was missing (HCO3- from the atmosphere). We have now accounted for this and updated the text to make the corrections done clearer. Values corrected in the text, figure 3, 4, 5, 6, and 7 as well as Table 1, 2 and 3. The above modifications do not change the story/interpretation of the paper, but did make a significant enough change to warrant updating rates, figures and tables.*

**The authors recalculate their rates in an attempt to account for anthropogenic impact on erosion/weathering. However, the equations used for this calculation are only shown in the discussion. They should be either stated in the methods, or the explanation of the anthropogenic disturbance correction should be removed from the methods and only mentioned in the discussion. After only reading the methods, and therefore not knowing the equations, I spent a considerable amount of time looking at the data table, and trying to figure out what was done.**

*Our apologize for the unclarity and the time spent figuring out what we did. We found the source of confusion. The anthropogenic corrected rates are presented in Table S12, which we referenced in the methods section. However, the calculations shown in Table S12 are not referred to again, or used, until the discussion section 5.1.2 where we do explain how the correction is applied. There is no need for us to reference table S12 in the methods, so we have removed this. The methods section reads now as:" Anthropogenic impacts on each catchment were evaluated using three different main approaches. First, the connectivity status index (CSI) for river systems was evaluated for each catchment, where values of 100% represent undisturbed rivers (Grill et al., 2019). Second, the human footprint index (HFI) was extracted with the highest impact to be 50 and the lowest 0 (Mu et al., 2022). This HFI represents the relative anthropogenic influence in each terrestrial biome and is represented as a percentage. Third, the % of different landcover such as artificial surfaces and constructions as well as cultivated area, vineyard or tree covers were extracted from the landcover map of Europe 2017 (Malinowski et al., 2020). The third metric used for anthropogenic impact is the % area of artificial constructions and surfaces. ".*

**I was surprised to see that the authors attempted to correct for the anthropogenic impact on erosion/weathering rates. As the authors state themselves, there is no established way of doing this and the effects of agriculture could be non-linear. The authors state several times how uncertain these correction with relatively arbitrary equations are. The**

**authors use these corrections to argue that the Neckar tributaries erode faster than the Danube tributaries even after correction. Wouldn't it be more straightforward to show boxplots or state the values of average HFI, cultivated area, CSI etc for Neckar and Danube side. If the distributions of human influence are similar in both drainage basins, one can make the argument that the difference in erosion/weathering is due to natural factors.**

*The average distributions of human influence are the following for the Neckar and the Danube side:*

*CSI: Neckar 99.16%; Danube 99.54% (similar in both catchments)*

*HFI: Neckar 29.59%; Danube 25.19% (different between the catchments)*

*Artificial area: Neckar 9.7%, Danube 5.2% (different between the catchments)*

*So the argument cannot be made that the correction would affect both sides of the Swabian Alb in the same way and the differences in rates are due to natural factors. Therefore, we decided to apply a correction to each rate for the specific catchment. However, as the review notes – we very cautiously do this and every time we mention this correction we have clear text saying how to do this correctly is unknown and that we are only working with an estimate here.*

**Response to reviewer comments RC 2**

In their revised manuscript entitled 'Spatiotemporal denudation rates of the Swabian Alb escarpment (Southwest Germany) dominated by anthropogenic impact, lithology, and base-level lowering', Schaller et al. analyse spatial and temporal patterns of weathering, erosion and denudation rates for the Swabian Alb. The authors have substantially modified the earlier version of the manuscript to take into account the reviewers' comments. Although the current version is a great improvement, I still have some comments, which are listed below.

**Methods**
An important comment from both reviewers related to the incomplete description of the methods, which made it difficult to follow the individual approaches presented in the study and hindered the reproduction of the analyses. In the revised version, the authors go into sufficient detail and readability has been improved. I have 4 comments here.

Firstly, it is still unclear how weathering and erosion rates are ultimately quantified using TSS and TDS. Section 3.2 gives a general explanation of which parameters are used in the calculation, but the equations themselves are not given. I assume that the equations shown in Figure 1 on the left are used. I suggest either referring to Figure 1 here or listing the equations in Section 3.2 and possibly removing them from Figure 1 (see also comment below).

*L291-292: Reference to Fig. 1 has been added to the text to clarify how the weathering rate is calculated. In addition, we refer to Table S2 (formerly Table S6A) to improve the clarity of the weathering rate calculation.*

Secondly, I appreciate the fact that the authors have now introduced three new proxies to quantify the anthropogenic influence on the catchment areas (lines 264-277). However, it is not clear from the description how the values were derived and what these parameters actually mean. Were the values calculated by the authors or taken from existing data sets? What does the connectivity status index (CSI) describe, lateral or downstream connectivity or both? And if the
human footprint index (HFI) is given in % and the highest values found in the study region are ~50 %, why are 50 % and not 100 % used in the calculation of anthropogenic impacts (Section 5.1.2)?

*There was some unclarity in the method section as already stated by reviewer 1. We hope that the adjustments made to improve the clarity of the manuscript for the reader. We further reference back in the discussion section to the method section where CSI, HFI, and % area of artificial constructions are described and references provided.*

Thirdly, I appreciate that the authors have tried to take into account the anthropogenic influence on erosion, weathering and denudation rates. However, I am a little concerned about the approach used. Although the authors clearly point out that the approach should be taken with caution as there is no standard procedure, I think the approach presented corrects in the wrong direction. Lines 465-465 state: '[...] many TSS values have declined by up to 50% in large German rivers (2 000 to 160 000 km2) in the last ~20 years (Hoffmann et al., 2023). Such a decrease in TSS is usually observed in the northern hemisphere due to dams (Dethier et al., 2022).' This statement implies that human influence has led to a reduction in TSS and thus weathering rates. Natural/unmodified rates would therefore be higher. However, the weighting of the anthropogenically corrected rates proposed here (Section 5.1.2) leads to a reduction in the natural

**rates compared to the measured human-influenced rates. For example, the greater the human influence, the lower the ratio of CSImean/100 and the lower the weighted natural rates. In view of the above statement, shouldn't the correction go in the other direction and instead increase the natural rates with increased human influence?**

*We appreciate the reviewer's thoughtful engagement with our correction approach and agree that the 'direction' of correction for anthropogenic impact is not straightforward. As noted, human activities can have contrasting effects on sediment and solute yields depending on the specific processes at play. While damming and different forms of river regulation have indeed led to widespread reductions in total suspended sediment (TSS) in many large rivers, our study focuses on smaller, upland catchments where anthropogenic activities such as agriculture, urbanization, deforestation, and mining are more likely to **enhance** rather than suppress erosion and weathering rates. In this context, we assume that human impact tends to elevate measured denudation, erosion, and weathering rates beyond what would occur under more natural, less disturbed conditions. Consequently, we applied a correction that **reduces** the observed (human-influenced) rates proportionally to the magnitude of anthropogenic disturbance.*

*We fully acknowledge, however, that this assumption may not hold in all cases — particularly in highly regulated lowland rivers or where sediment trapping dominates. We have revised Section 5.1.2 to clarify the rationale behind our chosen correction direction and have added a stronger caveat that this approach may not capture all facets of human impact. We also highlight in the revised text that future studies should consider process-based corrections that can distinguish between erosive and sediment-trapping anthropogenic effects.*

*In summary, while the reviewer raises a valid point, we believe that for the specific geomorphic setting of our study — relatively small, mountainous catchments with evidence of anthropogenically elevated sediment and solute fluxes — our correction logic is appropriate. Nonetheless, we agree that this issue deserves more attention.*

**Fourth, I am still concerned about the approach to examining the relationship between erosion, weathering or denudation rates with topographic, climatic or biotic average catchment parameters (lines 342-345). Although I understand why the authors favour linear regressions over polynomial regressions, several previous studies have shown non-linear relationships between these rates and catchment average parameters. Therefore, instead of calculating the Pearson correlation coefficient, which assumes linear relationships, I suggest calculating the Spearman rank correlation coefficient instead. This measures the strength and direction of two variables (monotonically increasing or decreasing), but does not assume linearity. I think this approach makes far fewer assumptions about the underlying relationships, but still provides a metric similar to the one presented in the study.**

*L342-345: The reviewer is right that the Spearman rank correlation coefficient does not assume a certain relationship. However, it should be noted that both metrics (Pearson vs. Spearman) produce a similar result in terms of the direction of the correlation, or no correlation (e.g., positively, negatively, or no correlation). To address the reviewer's concern we have calculated the Spearman correlation for reference and show it in the table below, and also show the difference between the two approaches. It should be noted that a) the direction of the correlation calculated is the same in almost all cases, and b) the correlations are almost all weakly correlated (e.g., << 0.5 or -0.5) and support our original interpretation of weak to moderate correlations between denudation rates and the various metrics analysed. As this additional analysis does not change our interpretations or conclusions, we have left our analysis in the text to discuss the Pearson correlation relationship. We do this because it provides an evaluation of the potential linearity in the relationships analysed, which we find useful to report.*

| Metric | Pearson all data | | | Spearman all data | | | Difference Spearman-Pearson | | |
|---|---|---|---|---|---|---|---|---|---|
| | $W_{corr.}$ | $E_{corr.}$ | $D_{corr.}$ | $W_{corr.}$ | $E_{corr.}$ | $D_{corr.}$ | $W_{corr.}$ | $E_{corr.}$ | $D_{corr.}$ |
| | n=43 | n=43 | n=43 | n=43 | n=43 | n=43 | n=43 | n=43 | n=43 |
| Catchment area | 0.03 | 0.05 | 0.07 | -0.01 | 0.41 | 0.11 | -0.04 | 0.36 | 0.04 |
| Mean elevation | -0.17 | -0.15 | -0.27 | -0.16 | -0.32 | -0.30 | 0.02 | -0.17 | -0.03 |
| Max. relief | 0.07 | 0.44 | 0.41 | 0.27 | 0.71 | 0.46 | 0.20 | 0.28 | 0.05 |
| Local relief (1000m) | -0.09 | 0.22 | 0.09 | 0.09 | 0.26 | 0.11 | 0.18 | 0.03 | 0.02 |
| Trunk mean_$k_{sn}$ | -0.22 | 0.20 | 0.01 | -0.13 | 0.30 | 0.03 | 0.08 | 0.11 | 0.03 |
| Slope | -0.07 | 0.21 | 0.09 | 0.13 | 0.32 | 0.17 | 0.20 | 0.11 | 0.08 |
| Mean annual precipitation | -0.40 | -0.24 | -0.40 | -0.41 | -0.05 | -0.38 | -0.01 | 0.19 | 0.02 |
| Mean annual temperature | 0.22 | 0.17 | 0.31 | 0.31 | 0.30 | 0.42 | 0.09 | 0.13 | 0.12 |
| NDVI (Vegetation cover) | -0.04 | -0.02 | -0.03 | 0.02 | 0.27 | 0.10 | 0.06 | 0.29 | 0.13 |
| Soil depth | 0.20 | -0.07 | 0.11 | 0.10 | 0.02 | 0.13 | -0.10 | 0.09 | 0.01 |
| Connectivity Status Index | -0.20 | -0.55 | -0.52 | -0.38 | -0.41 | -0.50 | -0.18 | 0.14 | 0.02 |
| Human Footprint Index | 0.49 | 0.26 | 0.52 | 0.52 | 0.25 | 0.53 | 0.02 | -0.01 | 0.01 |
| Artificial constructions | 0.52 | 0.52 | 0.66 | 0.55 | 0.38 | 0.54 | 0.03 | -0.14 | -0.12 |
| Cultivated area/vineyards | 0.22 | -0.07 | 0.11 | 0.18 | -0.10 | 0.16 | -0.04 | -0.03 | 0.06 |
| LowerTriassic | -0.43 | -0.11 | -0.32 | -0.40 | -0.03 | -0.35 | 0.04 | 0.07 | -0.04 |
| MiddleTriassic | 0.13 | -0.10 | 0.01 | 0.26 | 0.16 | 0.22 | 0.13 | 0.25 | 0.21 |
| UpperTriassic | 0.48 | 0.13 | 0.35 | 0.72 | 0.53 | 0.67 | 0.24 | 0.41 | 0.32 |
| LowerJurassic | 0.62 | 0.44 | 0.62 | 0.65 | 0.71 | 0.75 | 0.02 | 0.27 | 0.13 |
| MiddleJurassic | 0.41 | 0.33 | 0.43 | 0.52 | 0.45 | 0.55 | 0.11 | 0.12 | 0.12 |
| UpperJurassic | -0.31 | -0.14 | -0.28 | -0.12 | -0.12 | -0.11 | 0.19 | 0.02 | 0.17 |
| Tertiary | -0.27 | -0.18 | -0.26 | -0.32 | -0.14 | -0.31 | -0.05 | 0.04 | -0.04 |
| Quaternary | 0.10 | 0.09 | 0.15 | 0.33 | 0.22 | 0.38 | 0.23 | 0.13 | 0.23 |

**Structure**
**The readability of the manuscript has been significantly improved compared to the previous version. And I understand that there are different writing and organising styles for a paper. However, I still find the introduction quite complicated, especially the two paragraphs on how denudation, erosion and weathering rates are quantified on different time scales (lines 62-95, Fig. 1). As mentioned earlier, after reading this introduction, I would expect a study that focuses specifically on bridging the gap between denudation rates at different time scales. Instead, much of the results and discussion focus on explaining the spatial variability of denudation rates and identifying parameters that control these rates. This part of the study is not justified in the introduction. It is also not clear from the introduction why the horizontal retreat rates are calculated. Therefore, I would again suggest that the structure of the introduction be better aligned with the content of the rest of the manuscript.**
*We struggle to understand why the reviewer interprets this from the introduction. The other reviewer hasn't highlighted this, and none of the co-authors see the manuscript as an attempt to bridge between timescales. The words "bridge" or "bridging" are never used in the manuscript. The current text in the introduction simply highlights that conventional approaches used are sensitive to different time scales (i.e., truth in advertising). To accommodate the reviewer's confusion, we have added a sentence in the introduction stating that there is currently no way to bridge timescales between these different metrics, but we mention it there is a timescale sensitivity in the methods used. We've also added a sentence in the introduction (last paragraph) why horizontal retreat rates are calculated.*

**Technical corrections**

**• Sometimes abbreviations like W or E are not in italics (e.g., lines 172, 279, and more).**
*Further abbreviations not in italics were found and corrected for.*

**• Lines 184-185: "**
**…drainage systems of the Neckar River draining Northward the Rhine River in the Northwest and the Danube River draining to the Southeast." The sentence needs to be grammatically corrected.**
*L1884-185: The sentence reads now as: "… the Neckar River draining Northward into the Rhine River in the Northwest and the Danube River draining to the Southeast."*

**• Table 2: What is meant by direction here, the aspect in degrees?**
*Yes, the aspect is meant. Direction has been replaced in Table 2 and Table S14.*

**• Lines 411-412: Remove italics.**
*L411-412: Italics have been removed. Thank you.*

**• Table 3 caption: "Correlations between corrected rates and mean… " What kind of correlations are reported here, what parameters are given in the table? I assume it is the Pearson correlation coefficient?**
*Yes, correct. Pearsons correlation coefficient has been added to the table caption:*

**• Line 473: A 'be' is missing here.**
*L473: The missing "be" has been added.*

**• Line 483: The closing parenthesis after 'Footprint Index' is missing.**
L483: Missing *part of parenthesis has been added.*

**• Line 688: Remove single letters in the sentence.**
*L688: Singel letters have been removed.*

---

## Author Response (AR3)

**Public justification (visible to the public if the article is accepted and published)**:
Dear authors,

Thank you very much for your re-revised manuscript and response to the reviewers. I am satisfied that you have addressed the reviewers' concerns, and note that the issue of how to correct denudation rates for anthropogenic impact is a difficult one, which in general needs more attention in the literature, but I appreciate your attempts at it here.

One minor point that could improve your justification of the anthropogenic corrections, and address Reviewer 2's concern about the direction of the correction (i.e. whether anthropogenic modification enhances or reduces denudation rates), would be to link back to the discussion of the direction of the correlation between the anthropogenic metrics and denudation rates. The positive correlation between human footprint index and artificial constructions with denudation rates, and the negative correlation with CSI, when all data are considered (Table 3) supports your argument that natural rates would be lower than the anthropogenically-influenced ones, rather than higher as Reviewer 2 suggested. I think you could make this point in Section 5.1.2 to strengthen your argument here.

Best wishes,
Fiona

Dear Fiona,
Thank you very much for your input and being the associate editor for this publication.

In order to further improve the manuscript, we added to Section 5.1.2 the following sentence: "The positive correlation between HFI and artificial constructions with denudation rates, and the negative correlation with CSI, when all data are considered (Table 3), supports the argument that natural rates are lower than the anthropogenically-influenced ones. We acknowledge, however, that not all human impacts increase erosion rates; for instance, dam construction may reduce downstream sediment transport and thus lower physical erosion, in which case the adjustment for applying a correction to denudation rates should be reversed."

In addition, we made the supplementary data publicly available through Zenodo.